# Sat3DGen: Comprehensive Street-Level 3D Scene Generation from Single Satellite Image

**Ming Qian[1],     Zimin Xia[2],     Changkun Liu[3],     Shuailei Ma[4],**
**Wen Wang[5],   Zeran Ke[1],   Bin Tan[6],   Hang Zhang[7],   Gui-Song Xia[1]***

[1] LIESMARS & School of Artificial Intelligence, Wuhan University [2] EPFL
[3] HKUST [4] Northeastern University [5] Zhejiang University [6] Ant Group [7] Amap, Alibaba Group

## Abstract

Generating a street-level 3D scene from a single satellite image is a crucial yet challenging task. Current methods present a stark trade-off: geometry-colorization models achieve high geometric fidelity but are typically building-focused and lack semantic diversity. In contrast, proxy-based models use feed-forward image-to-3D frameworks to generate holistic scenes by jointly learning geometry and texture, a process that yields rich content but coarse and unstable geometry. We attribute these geometric failures to the extreme viewpoint gap and sparse, inconsistent supervision inherent in satellite-to-street data. We introduce Sat3DGen to address these fundamental challenges, which embodies a geometry-first methodology. This methodology enhances the feed-forward paradigm by integrating novel geometric constraints with a perspective-view training strategy, explicitly countering the primary sources of geometric error. This geometry-centric strategy yields a dramatic leap in both 3D accuracy and photorealism. For validation, we first constructed a new benchmark by pairing the VIGOR-OOD test set with high-resolution DSM data. On this benchmark, our method improves geometric RMSE from 6.76m to 5.20m. Crucially, this geometric leap also boosts photorealism, reducing the Fréchet Inception Distance (FID) from ∼40 to 19 against the leading method, Sat2Density++, despite using no extra tailored image-quality modules. We demonstrate the versatility of our high-quality 3D assets through diverse downstream applications, including semantic-map-to-3D synthesis, multi-camera video generation, large-scale meshing, and unsupervised single-image Digital Surface Model (DSM) estimation. The code has been released on `https://github.com/qianmingduowan/Sat3DGen`.

## 1 Introduction

Street-level 3D scenes are useful for mapping, robotics, simulation, and media creation (Workman et al., 2017; Toker et al., 2021; Xie et al., 2024; Zhou et al., 2020; Shi et al., 2022; Li et al., 2024a). Ground-level capture is costly and uneven across regions (Anguelov et al., 2010), whereas satellite imagery offers wide coverage, low cost, and frequent updates (Campbell & Wynne, 2011). These characteristics motivate the generation of street-level 3D from overhead satellite images for large-scale, long-term applications. Our goal is to generate a 3D scene that faithfully preserves the semantics and appearance of an input satellite image and that can be rendered for street-view images and videos.

Existing methods for generating 3D from a single satellite image fall into two categories: *3D geometry colorization* (Hua et al., 2025; Li et al., 2024b) and *3D proxy for image rendering* (Qian et al., 2023; 2026). *3D geometry colorization* follows a two-stage pipeline to predict and then texture 3D building geometry. While producing clean building models, these methods fail to capture non-building elements (e.g., zebra crossings, trees), resulting in outputs weakly consistent with the input satellite image (Fig. 1 (a,b)).[1] Extending them beyond buildings would require fine-grained geometry labels for many classes, which are scarce. *3D proxy for image rendering* uses tailored *feed-forward image-to-3D frameworks* (Hong et al., 2024; Xiang et al., 2025; Yu et al., 2021; Zhang et al., 2025) to learn

---

*Corresponding author

[1]As of the submission deadline, the official implementations of Sat2Scene and Sat2City had not been fully released. We therefore report the 3D results shown in their papers and project pages.

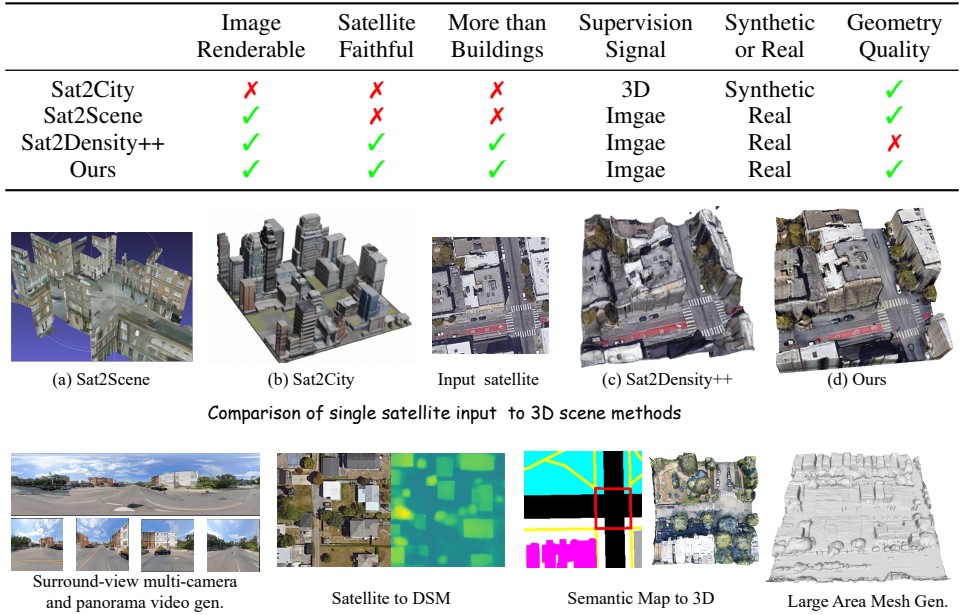

| | Image Renderable | Satellite Faithful | More than Buildings | Supervision Signal | Synthetic or Real | Geometry Quality |
|---|---|---|---|---|---|---|
| Sat2City | ✗ | ✗ | ✗ | 3D | Synthetic | ✓ |
| Sat2Scene | ✓ | ✗ | ✗ | Imgae | Real | ✓ |
| Sat2Density++ | ✓ | ✓ | ✓ | Imgae | Real | ✗ |
| Ours | ✓ | ✓ | ✓ | Imgae | Real | ✓ |

(a) Sat2Scene  (b) Sat2City  Input satellite  (c) Sat2Density++  (d) Ours

Comparison of single satellite input to 3D scene methods

Surround-view multi-camera and panorama video gen.  Satellite to DSM  Semantic Map to 3D  Large Area Mesh Gen.

High-quality Renderable 3D generation from satellite image support multiple applications...

Figure 1: Comparison of 3D scene generation methods (top: attribute table; bottom: visual results). Given an input satellite image, (a) Sat2Scene and (b) Sat2City generate only shells of buildings and roads and miss non-building semantics; (c) Sat2Density++ and (d) Ours are faithful to satellite semantics and appearance, but Sat2Density++ is heavily distorted, whereas our Sat3DGen yields a more structured, higher-quality 3D representation.

a coarse, differentiable 3D proxy via joint optimization under 2D supervision. These methods are semantically faithful but yield poor geometry: boundaries are degraded, roofs are unrealistic, and floating artifacts are common (Fig. 1 (c)).

Our goal requires preserving the rich semantics of the input satellite image, making the proxy-based paradigm more suitable than geometry colorization. Encouragingly, recent object-level *feed-forward image-to-3D works* (e.g., InstantMesh (Xu et al., 2024), LRM (Hong et al., 2024)) have demonstrated that high-quality 3D can be learned from 2D supervision alone. This suggests that the poor geometry of existing scene-level proxy methods is not a fundamental flaw of the paradigm. Instead, we hypothesize it stems from insufficient geometric constraints to handle the unique challenges of outdoor scenes. Specifically, the supervision from only one satellite patch and a few ground-level panoramas is extremely sparse. This sparsity, coupled with the extreme viewpoint gap, leaves rooftop geometry underconstrained and induces artifacts like holes and floaters on vertical facades. Additionally, a footprint mismatch between the satellite and street views often destabilizes the geometry at scene boundaries.

To solve these specific problems, we propose Sat3DGen, which embodies a holistic, geometry-first methodology. Our strategy is not to invent a new feedforward image-to-3D architecture from scratch, but to elevate a general framework by demonstrating how to effectively solve its core geometric failures. To enforce plausible vertical structures and suppress floating artifacts, we introduce a *Gravity-based Density Variation Loss*. To address boundary errors stemming from the footprint mismatch, a *Spatial Token* regularizes peripheral layouts. To resolve rooftop ambiguity, a *Monocular Relative-Depth Prior* constrains satellite-view depth. Furthermore, to mitigate the issue of sparse supervision, we employ *Perspective View Training*, jointly training on panoramas and their projected views to increase effective viewpoint coverage and photometric consistency.

In evaluation, this emphasis on geometry translates directly to substantial quantitative and perceptual gains. We first validate our geometric improvements against the leading method, Sat2Density++, on a new benchmark we constructed by pairing the VIGOR-OOD test set with 1-meter resolution DSM data. Sat3DGen achieves a geometric RMSE of 5.20m, a significant reduction from Sat2Density++'s 6.76m. Crucially, this leap in 3D accuracy directly fuels a dramatic improvement in photorealism. Even though it includes no components tailored to image quality, our framework reduces the Fréchet

Inception Distance (FID) on the VIGOR-OOD unseen-city split from Sat2Density++'s ∼40 to 19. The resulting assets support diverse downstream applications such as semantic-map-to-3D synthesis, surround-view video from satellite imagery, large-area mesh generation, and single-image Digital Surface Model (DSM) generation without ground-truth depth supervision.

## 2 RELATED WORKS

**Feed-Forward Image to 3D Works** has gained popularity for producing high-quality 3D assets. Recently, large reconstruction models (Hong et al., 2024; Tang et al., 2024; Xu et al., 2024; Xiang et al., 2025) have focused on generating object-level 3D assets, leveraging larger datasets, more refined annotations, and more substantial models to improve the quality of the generated assets, achieving impressive results. However, existing works primarily focus on object-level generation, presenting additional challenges when applying these models in outdoor scenes. In our work, we focus on generating high-quality, comprehensive street-level 3D from a single input satellite image, thereby naturally enhancing the quality of generated videos and supporting various applications.

**Single Satellite to Street-view Synthesis.** Early studies generate individual street-view images from a single satellite patch (Regmi & Borji, 2018; 2019; Toker et al., 2021; Shi et al., 2022; Lu et al., 2020; Tang et al., 2019), but they do not produce usable 3D or multi-view consistency. Later works synthesize street-view videos by learning a colored 3D asset from the satellite input (Li et al., 2021; 2024b; Qian et al., 2026). Geometry-colorization methods (Li et al., 2021; 2024b) often rely on height maps and vertical-facade assumptions, yielding building-centric scenes and missing non-building semantics such as roads, crosswalks, and trees. Our work builds on the proxy-based line and focuses on improving 3D quality under the same single-satellite input setting.

## 3 METHOD

As shown in Fig. 2, given a single overhead satellite image $I_{sat}$ and an optional global illumination feature input $f_{ill}$ used solely to control illumination when rendering street views, our model can synthesize a renderable 3D scene that (i) preserves the semantics and appearance of $I_{sat}$, (ii) supports high-fidelity satellite, perspective street-view, and panoramic rendering under controllable lighting, and (iii) can be exported as a mesh with Marching Cubes.

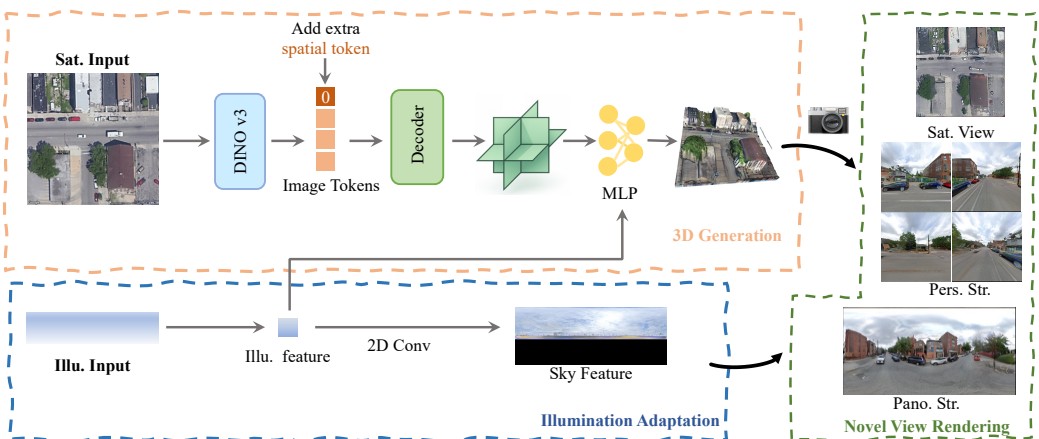

Figure 2: Diagram of the proposed Sat3DGen framework.

We adopt a feed-forward image-to-3D framework instantiated with a tri-plane NeRF (Chan et al., 2022) as a baseline. A frozen DINO-v3 encoder (Siméoni et al., 2025) maps $I_{sat}$ to a compact token grid, which is optionally padded with learnable spatial capacity at the periphery and then decoded into a high-resolution tri-plane feature field. A lightweight MLP predicts density and color features from tri-plane features for volumetric rendering. Besides, we follow the illumination-adaptive design in Qian et al. (2026) to mitigate the sky/illumination mismatch issue.

Beyond this backbone, we introduce three novel geometry-oriented components that substantially enhance performance and depart from prior work (Qian et al., 2026): a gravity-based density variation loss to favor gravity-aligned structures, a monocular relative-depth prior in satellite view to resolve rooftop ambiguity, and panoramic-to-perspective supervision to densify viewpoints. The remainder of this section details the backbone; the losses and supervision strategy are presented in Section 3.4.

## 3.1 SATELLITE TO 3D GENERATION

Given a satellite image, our pipeline constructs a radiance field by encoding it into a 2D token grid with a frozen backbone, padding with spatial tokens to expand the effective scene extent, and decoding the tokens into tri-plane features.

**Satellite Encoder and Tokenization.** Following exsiting object-level feedforward image to 3D works, we use frozen pretrained VIT model as image encoder (Xu et al., 2024; Xiang et al., 2025). In practise, a frozen DINO-v3 ViT encoder (Siméoni et al., 2025) $\mathcal{E}_{\text{sat}}$ processes $I_{\text{sat}}$ into a 2D token grid:

$$\mathbf{F}_{\text{token}} = \mathcal{E}_{\text{sat}}(I_{\text{sat}}) \in \mathbb{R}^{H_t \times W_t \times C}, \tag{1}$$

with $H_t = W_t = 16$ and $C = 1024$ in all experiments. This token grid is the minimal scene-level latent that will be lifted into a 3D feature field.

**Spatial Tokens.** Street-view supervision often observes buildings and roads extending beyond the satellite crop, which induces boundary artifacts if the 3D field is constrained to the crop footprint. We therefore pad $\mathbf{F}_{\text{token}}$ with a border of $N$ zero-valued spatial tokens on each side:

$$\mathbf{F}_{\text{token\_pad}} = \text{PAD}_N(\mathbf{F}_{\text{token}}) \in \mathbb{R}^{(H_t+2N) \times (W_t+2N) \times C}, \quad N = 2. \tag{2}$$

With $H_t = W_t = 16$, padding yields $\mathbf{F}_{\text{token\_pad}} \in \mathbb{R}^{20 \times 20 \times 1024}$. Suppose the original scene cube spans $L$ meters per side (e.g., $L = 50\,\text{m}$). In that case, padding enlarges the effective cube to $L \cdot \left(1 + \frac{2N}{H_t}\right)$ (e.g., 62.5 m), providing degrees of freedom to accommodate peripheral content while stabilizing interior geometry.

**Tokens → Tri-Plane Features.** A lightweight VAE-style decoder (Esser et al., 2021) $\mathcal{D}$ upsamples tokens into a high-resolution tri-plane feature map with an upsampling factor $s = 16$:

$$\mathbf{F}_{\text{tri}} = \mathcal{D}(\mathbf{F}_{\text{token\_pad}}) \in \mathbb{R}^{\text{res}_{\text{tri}} \times \text{res}_{\text{tri}} \times 96}, \tag{3}$$

where $\text{res}_{\text{tri}} = 320$ when padding is used and 256 otherwise. Channels are reshaped into three orthogonal planes $(XY, XZ, YZ)$.

**Tri-Plane Sampling.** A 3D query point $\mathbf{x} \in \mathbb{R}^3$ within the normalized scene cube is orthographically projected onto each plane and bilinearly sampled to obtain features $\phi_{XY}(\mathbf{x}), \phi_{XZ}(\mathbf{x}), \phi_{YZ}(\mathbf{x})$. The three plane features are aggregated by elementwise summation to form the fused feature:

$$\mathbf{h}(\mathbf{x}) = \phi_{XY}(\mathbf{x}) + \phi_{XZ}(\mathbf{x}) + \phi_{YZ}(\mathbf{x}). \tag{4}$$

Then, a shallow MLP predicts density and color:

$$\sigma(\mathbf{x}), \mathbf{c}(\mathbf{x}, \mathbf{w}) = \text{MLP}(\mathbf{h}(\mathbf{x}), \mathbf{w}), \tag{5}$$

where $\sigma(\mathbf{x})$ denotes the volume density, $\mathbf{c}(\mathbf{x}, \mathbf{w})$ is the radiance color conditioned on an illumination code $\mathbf{w}$, and $\mathbf{h}(\mathbf{x})$ is the fused tri-plane feature at location $\mathbf{x}$; the MLP uses a shared trunk with two output heads for density and color.

## 3.2 ILLUMINATION-ADAPTIVE RENDERING AND SKY GENERATION

**Global Illumination Code.** Following Sat2Density++, we extract a global illumination feature $f_{\text{ill}}$ from a real street-view image $I_{\text{ill}}$ in a statistical way (Qian et al., 2026), and then project to a style code $\mathbf{w}_{\text{ill}}$ with a light mlp:

$$\mathbf{w}_{\text{ill}} = \mathcal{E}_{\text{ill}}(I_{\text{ill}}). \tag{6}$$

During training, $f_{\text{ill}}$ is extracted from the groundtruth street-view panorama image to mitigate sky/illumination mismatch, and at test time, it enables lighting-controllable rendering.

**Sky Region Generation with Spherical Feature Maps.** To natively support perspective view rendering, the sky module must provide consistent appearances for arbitrary viewpoints. We achieve this by modeling the sky as a feature map on the sphere. A lightweight 2D convolutional decoder produces this sky feature map from $w_{\text{ill}}$:

$$\mathbf{F}_{\text{sky}} \;=\; \mathcal{G}_{\text{sky}}(w_{\text{ill}}) \;\in\; \mathbb{R}^{512 \times 512 \times c}, \tag{7}$$

where $c$ matches the renderer's feature channels. For any given ray with normalized direction $\mathbf{d} \in \mathbb{S}^2$, we convert its Cartesian coordinates to spherical angles $(\theta, \phi)$ and bilinearly sample $\mathbf{F}_{\text{sky}}$ to obtain the sky color feature $\mathbf{c}_{\text{sky}}(\mathbf{d})$. This design elegantly provides consistent sky features for both panoramic and perspective-view rendering.

## 3.3 VOLUMETRIC RENDERING AND OUTPUTS

**Ray Marching and Compositing.** For a camera ray $r(t) = \mathbf{o} + t\mathbf{d}$, $t \in [t_n, t_f]$, we sample points $\{\mathbf{x}_k\}$ with step $\delta_k$ and compute transmittance $T_k = \exp\left(-\sum_{j<k} \sigma(\mathbf{x}_j)\,\delta_j\right)$. The rendered color is

$$\mathbf{C}(r) \;=\; \sum_k T_k \left(1 - e^{-\sigma(\mathbf{x}_k)\delta_k}\right) \mathbf{c}(\mathbf{x}_k, w_{\text{ill}}) \;+\; T_{\text{out}}\, \mathbf{c}_{\text{sky}}(\mathbf{d}), \tag{8}$$

where $T_{\text{out}}$ is the remaining transmittance upon exiting the volume. The same renderer supports perspective and spherical cameras; the latter yields full panoramas.

**Renderable Views and Mesh Export.** Our model can render (i) satellite views, (ii) perspective street-view images at arbitrary camera poses, and (iii) panoramic street views. For asset export, we evaluate $\sigma$ on a dense grid and run marching cubes with a fixed isovalue $\tau$ to obtain a watertight mesh. The sky branch is excluded from meshing.

## 3.4 LOSS FUNCTIONS.

**Gravity-based Density Variation Loss**. Outdoor scenes reconstructed from sparse views often exhibit geometric artifacts like floating debris and hollow grounds. To mitigate these issues, we introduce a regularizer based on a simple design principle: volumetric density should generally be non-increasing with altitude. The design of this regularizer is motivated by the physical effect of gravity. To translate this concept into the NeRF framework, we leverage the volume density $\sigma$. In NeRF, $\sigma$ measures light obstruction, making it a natural proxy for physical matter, given that outdoor scenes are predominantly composed of opaque surfaces like terrain, rocks, and tree trunks. Following the intuition that gravity causes matter to accumulate at lower elevations, we establish our principle: $\sigma$ should generally be non-increasing with altitude. This is consistent with real-world observations; for instance, solid ground and tree trunks are typically found at lower altitudes, while higher altitudes often contain sparser structures like leafy canopies or simply open air. Grounding our regularizer in this physical intuition helps the model learn more plausible geometry.

Specifically, we sample a 3D point $\mathbf{x} \in \mathbb{R}^3$ and a corresponding point $\mathbf{x}' = \mathbf{x} + \delta\mathbf{z}$ at a slightly higher altitude, where $\delta\mathbf{z}$ is a small displacement vector purely in the upward (anti-gravity) direction. We then penalize cases where the density at the higher point $\mathbf{x}'$ is significantly greater than the density at the lower point $\mathbf{x}$. This is enforced by minimizing the following loss:

$$\mathcal{L}_{\text{grav}} = \mathbb{E}_{\mathbf{x},\delta\mathbf{z}} \left[\text{ReLU}(\sigma(\mathbf{x} + \delta\mathbf{z}) - \sigma(\mathbf{x}) - \epsilon)\right], \tag{9}$$

where the slack variable $\epsilon$ (set to 1 in our experiments) provides a soft constraint, allowing for legitimate hollow or overhanging structures such as tree canopies, arched roofs, and bridges. This loss effectively suppresses floating artifacts and fills baseless cavities while preserving realistic sparsity under overhangs.

**Satellite-View Depth Regularization**. Each scene provides one bird's-eye satellite image and only a few street-view observations; rooftops lack multi-view photometric supervision and tend to be irregular. We therefore impose a relative depth prior in the satellite view using pseudo labels from Depth Anything v2 (Yang et al., 2024). Let $D^*$ be the pseudo relative depth for the satellite camera and $\hat{D}$ the rendered depth from our field. We adopt a scale-and-shift invariant MiDaS-style loss (Ranftl et al., 2022):

$$\mathcal{L}_{\text{depth}} \;=\; \frac{1}{N} \sum_p \left|s\,\hat{D}(p) + t - D^*(p)\right| \;+\; \lambda_\nabla \frac{1}{N} \sum_p \left\|\nabla\!\left(s\,\hat{D}(p) + t\right) - \nabla D^*(p)\right\|_1, \tag{10}$$

where $(s, t)$ are optimal scale and shift estimated per image by least squares, $N$ is the number of valid pixels, and $\nabla$ denotes spatial gradients. This encourages consistent depth ordering and smooth rooftops without requiring metric depth.

**Photometric Reconstruction and Adversarial Loss**. We supervise three rendered view types: satellite views, panoramic street views, and perspective crops projected from panoramas. Let $\hat{I}_i$ be a rendered image and $I_i^{\text{gt}}$ the corresponding ground truth. The photometric objective combines per-pixel reconstruction with perceptual similarity, and we add an adversarial term to mitigate blur from pure regression in complex outdoor scenes:

$$\mathcal{L}_{\text{RGB}} = \sum_i \left\| \hat{I}_i - I_i^{\text{gt}} \right\|_2^2 + \lambda_{\text{lpips}} \sum_i \mathcal{L}_{\text{LPIPS}}\left( \hat{I}_i, I_i^{\text{gt}} \right) + \lambda_{\text{GAN}} \sum_i \mathcal{L}_{\text{GAN}}(\hat{I}_i), \qquad (11)$$

where $\mathcal{L}_{\text{LPIPS}}$ is the perceptual loss and $\mathcal{L}_{\text{GAN}}$ follows the StyleGAN2 hinge objective (Karras et al., 2020) for realism. In practice, the index $i$ ranges over satellite, panorama, and perspective supervision views rendered during training.

**Sky Losses: Opacity BCE and Masked Sky L1**. To disentangle the sky from the 3D scene and improve sky quality, we use two complementary losses on panoramic street views.

Let $M_{\text{sky}} \in \{0, 1\}^{H \times W}$ be the pseudo binary sky mask of a panorama (1 for sky), which is generated by the off-the-shelf model (Zhang et al., 2022), and let $T_{\text{out}} \in [0, 1]^{H \times W}$ be the residual transmittance per pixel from volumetric rendering (interpreted as the fraction attributed to the sky background after alpha compositing). We apply a binary cross-entropy:

$$\mathcal{L}_{\text{sky-op}} = \mathcal{L}_{\text{BCE}}(T_{\text{out}}, M_{\text{sky}}). \qquad (12)$$

Denote the rendered panorama $\hat{I}_{\text{pano}}$ and the ground-truth panorama $I_{\text{pano}}^{\text{gt}}$. We enforce color fidelity on sky pixels only:

$$\mathcal{L}_{\text{sky-L1}} = \frac{1}{\sum M_{\text{sky}}} \sum_p M_{\text{sky}}(p) \left\| \hat{I}_{\text{pano}}(p) - I_{\text{pano}}^{\text{gt}}(p) \right\|_1. \qquad (13)$$

**Overall Objective**. The full training objective is a weighted sum of the above terms:

$$\mathcal{L}_{\text{total}} = \lambda_{\text{rgb}} \mathcal{L}_{\text{RGB}} + \lambda_{\text{grav}} \mathcal{L}_{\text{grav}} + \lambda_{\text{sky-op}} \mathcal{L}_{\text{sky-op}} + \lambda_{\text{sky-L1}} \mathcal{L}_{\text{sky-L1}} + \lambda_{\text{depth}} \mathcal{L}_{\text{depth}}, \qquad (14)$$

where weights $\lambda_{\cdot}$ are hyperparameters.

## 4 EXPERIMENTS

**Datasets and Splits.** We train on GPS-matched satellite–ground image pairs. Training uses three cities (Chicago, New York, San Francisco) in the VIGOR dataset (Zhu et al., 2021), and out-of-domain (OOD) testing uses the held-out city Seattle (VIGOR-OOD). VIGOR provides multiple street-view panoramas per satellite tile together with relative camera poses; the satellite zoom level is fixed at 20, yielding near-constant ground sampling distance per pixel. In total, we use 78,188 pairs for training and 11,875 pairs for quantitative evaluation on VIGOR. More details, data preparation, and statistics are provided in Appendix F.

**Implementation Details.** We resize satellite images to $256 \times 256$ as input, and the generated triplane features have dimensions of $320 \times 320 \times 32 \times 3$. For fair comparison, the generated panorama images are shaped $512 \times 128$, and the perspective images are $256 \times 256$. The training process is conducted on 8 NVIDIA H20 GPUs with a batch size of 32, comprising 600,000 iterations for the training phase. More implementation details can be seen in the supplementary materials.

**3D Comparision.** We compare our 3D results with Sat2Scene (Li et al., 2024b), Sat2City (Hua et al., 2025) and Sat2Density++ (Qian et al., 2026). The colored meshes are generated by the Marching Cubes algorithm for Sat2Density++ and ours. Since there are no ground truth 3D assets available to evaluate the reconstruction quality, we can only perform qualitative comparisons, as shown on Fig. 1, Fig. 3, Fig. 4 (b), and Fig. 6. We observe consistent improvements in geometric plausibility

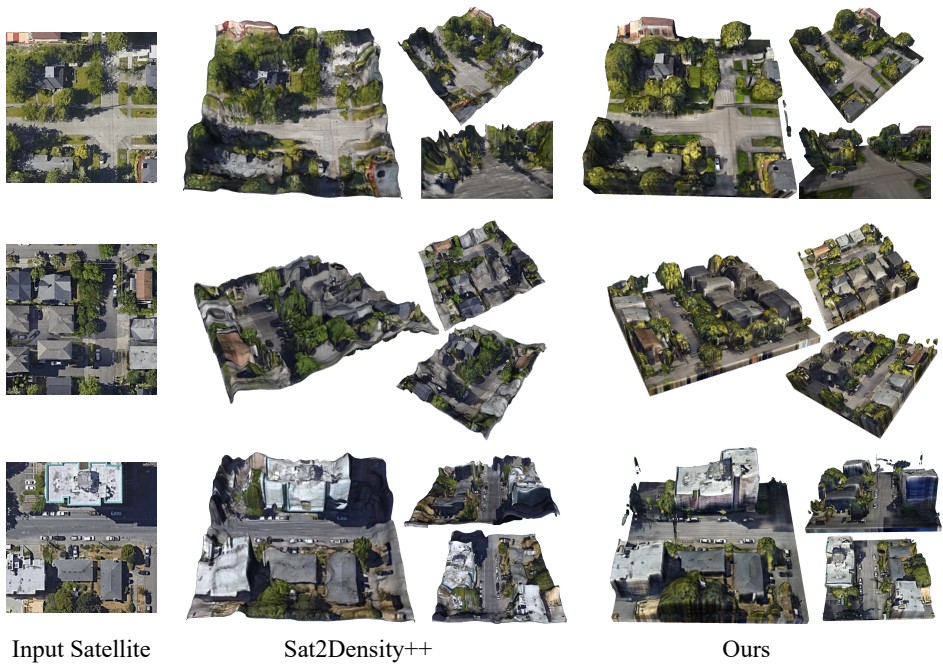

Figure 3: The comparison of generation 3D between Sat2Density++ (Qian et al., 2026) and our model on the VIGOR-OOD test set (Zhu et al., 2021).

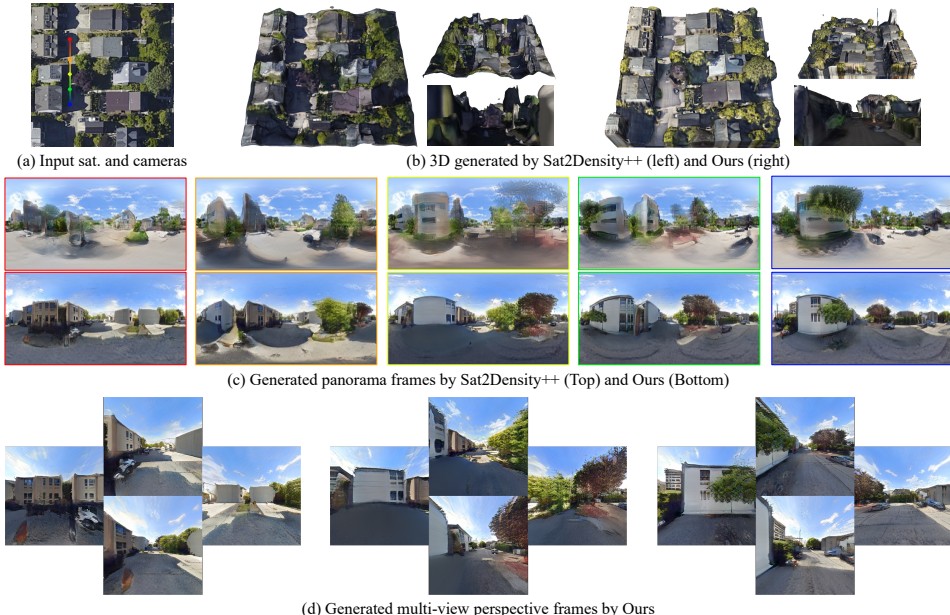

Figure 4: Visual results of generated mesh (b), panorama videos (c), and multi-view perspective video (d) from a single satellite image input and camera trajectories (rainbow line) (a). **The full video can be seen in the supplemental materials.**

and semantic faithfulness across diverse urban layouts. Compared with Sat2Scene and Sat2City, which mainly texture simplified building blocks and leave non-building regions weakly modeled, our reconstructions better preserve road markings, crosswalks, medians, tree belts, and sidewalks that are visible in the satellite input(Fig. 1. Relative to Sat2Density++, although both adopt a *feed-forward*

Table 1: Quantitative results of street-view comparison on the test set of VIGOR-OOD. **Bold** indicates the best results, while underlined text represents the second-best results.

| Method | Realism Evaluation | | Semantic | Structure | Pixel | Perceptual Similarity | |
|--------|------|------|------|------|------|------|------|
| | FID↓ | KID↓ | DINO↑ | SSIM↑ | PSNR↑ | $P_{alex}$ ↓ | $P_{squeeze}$ ↓ |
| ControlNet | 23.6 | / | / | 0.34 | 12.02 | 0.46 | 0.34 |
| ControlS2S | 28.0 | / | / | **0.42** | **13.80** | 0.38 | **0.27** |
| Sat2Density | 85.6 | 0.079 | 0.451 | 0.32 | 12.48 | 0.45 | 0.37 |
| Sat2Density++ | 40.8 | 0.035 | 0.465 | 0.34 | 12.51 | 0.44 | 0.34 |
| Canonical Image-to-3D | 35.6 | 0.030 | 0.479 | 0.35 | 12.63 | 0.42 | 0.32 |
| Ours | **19.2** | **0.014** | **0.525** | 0.37 | 12.83 | **0.38** | 0.30 |

*image-to-3D framework*, our method jointly integrates several lightweight components to improve geometry learning at street level under sparse, cross-view supervision. Taken together, these design choices strengthen scene layout near the satellite patch boundary, bias the volumetric field toward gravity-aligned structures, and inject rooftop depth cues from the overhead view, while increasing effective viewpoint coverage via panorama-to-perspective supervision. The resulting reconstructions exhibit more coherent ground planes and periphery geometry, with fewer torn edges and warped borders across the tile extent. Rooftops and building bases become geometrically plausible: roofs avoid bubbling or sagging, flat roofs remain planar, pitched roofs retain credible tilt, and facades connect cleanly to the ground (Fig. 1, Fig. 3, and Fig. 6).

**Image and Video Comparison.** We provide quantitative and qualitative comparisons. The qualitative comparison can be seen on Fig. 4, and more video comparisons are provided in the supplementary ZIP archive. The quantitative comparison is shown on Table 1.

Quantitative comparison. We follow prior work (Qian et al., 2026; Ze et al., 2025) for evaluation. We report Fréchet Inception Distance (FID) (Heusel et al., 2017) and Kernel Inception Distance (KID) (Bińkowski et al., 2018) to quantify distributional similarity between generated and real image sets, reflecting realism and coverage. Semantic alignment is assessed using a DINO-based feature similarity following (Qian et al., 2026). Pixel-level fidelity and structural similarity are evaluated with PSNR and SSIM, and perceptual similarity is measured with LPIPS (Zhang et al., 2018) using AlexNet and SqueezeNet backbones, denoted $P_{alex}$ (Krizhevsky et al., 2012) and $P_{squeeze}$ (Iandola et al., 2016). Given that our task is an input-view conditioned novel view generation problem with a very large viewpoint gap, pixel-level correspondence to any single real image is inherently brittle due to occlusions, parallax, and minor pose or scene changes. The primary desiderata are photorealism and semantic faithfulness rather than exact pixel matching. We therefore treat FID, KID, and DINO-based semantic similarity as primary metrics, and report PSNR, SSIM, and LPIPS for completeness.

We compare our model with Sat2Density (Qian et al., 2023), Sat2Density++ (Qian et al., 2026), the diffusion-based image generation model ControlNet (Zhang et al., 2023), and ControlS2S (Ze et al., 2025). We also provide results from a *feed-forward image-to-3D* model (denoted as "Canonical Image-to-3D" in Table 1), which removes the proposed spatial token module, gravity-based density variation loss, perspective training strategy, $\mathcal{L}_{grav}$, and $\mathcal{L}_{depth}$ proposed in our method. For Sat2Density, Sat2Density++, and our model, we pair each satellite image with a randomly selected global illumination feature input from the training set for a fair comparison. For ControlNet and ControlS2S, we use the results reported in the ControlS2S paper. It is worth noting that in the ControlS2S work, they divided the training and testing sets within each city. In contrast, our approach trains on three cities and conducts out-of-domain testing on an additional city, Seattle. This out-of-domain testing is significantly more challenging.

As shown in Table 1, our method leads in FID, KID, and DINO. Compared with the Canonical Image-to-3D, we only add our 3D optimization modules, yet the rendered images improve a lot. When compared with diffusion image generation models, our model also achieves lower FID and KID. This is because we learn a high-quality, view-consistent 3D representation that handles the large aerial-to-ground viewpoint change and produces more realistic and semantically correct images.

Qualitative Comparison. We provide a qualitative comparison in Fig. 4 (c), and more video comparisons can be seen in the zip supplemental materials. We can render videos from a given satellite image and any street-view camera trajectory, as illustrated in Fig. 4 (c) We compared the panorama video

Table 2: Ablation results on the VIGOR-OOD test set. The first row removes all proposed key components: $\mathcal{L}_{dep}$, $\mathcal{L}_{grav}$, Spatial Tokens, and perspective training. The next three rows ablate each component individually under the same setting (without perspective training). "Base (Full model w/o perspective training)" enables $\mathcal{L}_{dep}$, $\mathcal{L}_{grav}$, and Spatial Tokens but omits perspective training. "Full model" enables all components, including perspective training.

| | DINOv3 | $\mathcal{L}_{dep}$ | $\mathcal{L}_{grav}$ | Sp. Tok. | Per. Train | FID↓ | KID$_{\times 100}$↓ | RMSE↓ |
|---|---|---|---|---|---|---|---|---|
| Canonical Image-to-3D | ✓ | | | | | 35.6 | 30.1 | 6.21 |
| Base w/o $\mathcal{L}_{dep}$ | ✓ | | ✓ | ✓ | | 23.7 | 18.4 | 5.75 |
| Base w/o $\mathcal{L}_{grav}$ | ✓ | ✓ | | ✓ | | 25.9 | 19.0 | 5.21 |
| Base w/o Spatial Tokens | ✓ | ✓ | ✓ | | | 24.8 | 18.1 | 5.64 |
| Base (Full w/o per. training) | ✓ | ✓ | ✓ | ✓ | | 21.6 | 16.2 | 5.23 |
| Full model | ✓ | ✓ | ✓ | ✓ | ✓ | **19.2** | **13.6** | **5.20** |

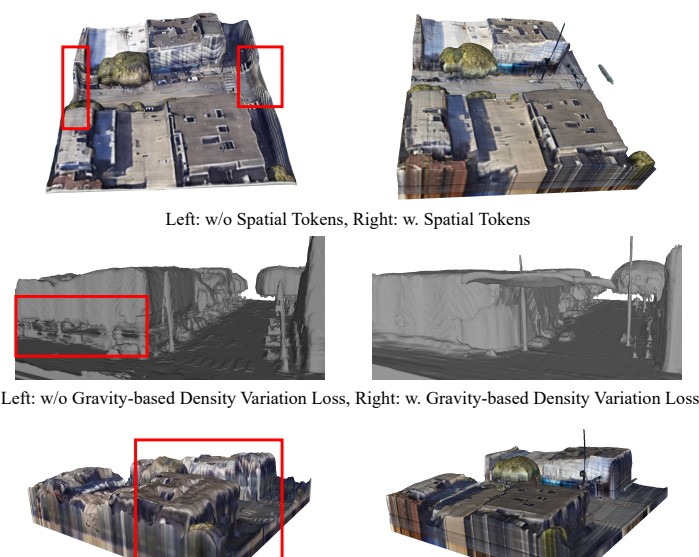

Left: w/o Spatial Tokens, Right: w. Spatial Tokens

Left: w/o Gravity-based Density Variation Loss, Right: w. Gravity-based Density Variation Loss

Left: w/o Satellite-view Depth Regularization, Right: w. Satellite-view Depth Regularization

Figure 5: Qualitative ablation on key modules.

results generated by Sat2Density++ and our model. Consistent with the conclusions drawn from the generated 3D assets, our model enhances the generated videos primarily by reducing artifacts and producing smoother edges around buildings and scene boundaries through the generation of higher-quality 3D representations.

**Geometric Comparison** To quantitatively evaluate the geometric accuracy of the generated 3D models, we compare the predicted satellite-view depth against the ground truth. The detailed procedure for preparing the ground truth DSM pairs for our VIGOR-OOD test set is provided in the Appendix Section D. As shown in Table 3, we adopt the standard evaluation metrics from satellite stereo literature (Gao et al., 2021), including Mean Absolute Error (MAE), Root Mean Square Error (RMSE), and the percentage of pixels with errors below 2.5m and 7.5m.

The results clearly demonstrate the effectiveness of our proposed methodology. The state-of-the-art method, Sat2Density++, achieves an RMSE of 6.76m. Our controlled baseline, Canonical Image-to-3D, which benefits from a stronger backbone, already improves upon this with an RMSE of 6.21m. Our full model, however, significantly outperforms both, establishing a new state-of-the-art with an RMSE of 5.20m and a MAE of 3.47m. Notably, our model reconstructs 62.69% of the surface with an error of less than 2.5m, a substantial improvement over Sat2Density++ (49.69%). This underscores the superior capability of our method in generating geometrically precise urban scenes. In addition to these quantitative results, we provide extensive qualitative comparisons of the rendered satellite-view

Table 3: Quantitative comparison for predicted DSM.

|                              | MAE↓ | RMSE↓ | $< 2.5\,\mathrm{m}\uparrow$ | $< 7.5\,\mathrm{m}\uparrow$ |
|------------------------------|------|-------|--------|--------|
| Sat2Density++                | 4.72 | 6.76  | 49.69  | 83.65  |
| Canonical Image-to-3D        | 4.23 | 6.21  | 52.73  | 84.54  |
| Base w/o $\mathcal{L}_{\mathrm{grav}}$ | 3.53 | 5.21  | 61.17  | 88.94  |
| Base w/o $\mathcal{L}_{\mathrm{dep}}$  | 3.82 | 5.75  | 59.88  | 87.04  |
| Base w/o Spatial Tokens      | 3.87 | 5.64  | 57.10  | 86.46  |
| Base (Full w/o per. training)| 3.52 | 5.23  | 61.97  | 88.52  |
| Ours (Full model)            | **3.47** | **5.20** | **62.69** | **88.68** |

depth and the groundtruth DSM in the Appendix Fig. 8, which visually corroborate our model's superior geometric fidelity.

**Ablation Study.** We conduct a comprehensive ablation study to validate our design choices, with quantitative results in Table 2, Table 3 and qualitative visualizations in Fig. 5. Starting from our Canonical Image-to-3D baseline (FID 35.6, RMSE 6.21m), integrating our core geometric priors ($\mathcal{L}_{\mathrm{dep}}$, $\mathcal{L}_{\mathrm{grav}}$, Spatial Tokens) synergistically boosts performance to an FID of 21.6. Among them, the gravity-based loss ($\mathcal{L}_{\mathrm{grav}}$) is most critical for photorealism, as its removal causes the largest FID degradation (to 25.9). Conversely, removing $\mathcal{L}_{\mathrm{dep}}$ or Spatial Tokens leads to more significant geometric errors (RMSE increases to 5.75m and 5.64m, respectively). Finally, adding perspective training achieves our best results across both photorealism (FID 19.2) and geometric accuracy (RMSE 5.20m).

These quantitative gains are visually corroborated in Fig. 5. As intended, Spatial Tokens regularize boundaries, $\mathcal{L}_{\mathrm{grav}}$ yields straighter facades and reduces floaters, and $\mathcal{L}_{\mathrm{dep}}$ corrects rooftop geometry. The consistent improvements across metrics and visuals confirm that each component is essential, targeting distinct aspects of the final high-fidelity reconstruction.

**Applications.** We show a few application examples in Fig. 1. Additional downstream results and implementation details are provided in Appendix Section B, including satellite-to-DSM (metric depth) conversion without ground-truth depth data supervision (Fig. 9), semantic map to 3D reconstruction (Fig. 11), large-area 3D mesh generation from a single large satellite patch (Fig. 10), and surround-view multi-camera video synthesis from satellite imagery (Section B.2).

## 5 CONCLUSION

In this work, we introduce Sat3DGen, a novel algorithm for generating a 3D outdoor scene from a satellite image. Our algorithm is trained exclusively on GPS-aligned satellite and panorama street view image data. By incorporating the novel approaches such as gravity-based Density Variation Loss and spatial tokens proposed in this paper, our experiments demonstrate that Sat3DGen effectively improves the quality of the generated 3D assets and enhances the quality of the generated videos. We hope that our approach inspires future research in 3D outdoor scene generation and that our model can effectively support downstream tasks such as digital Earth and scene simulation, thereby enhancing the efficiency and performance of outdoor 3D-related applications.

## ACKNOWLEDGEMENTS

This work was supported by NSFC under Grant 52325111.

## ETHICS STATEMENT

Our work builds on a publicly available dataset (VIGOR (Zhu et al., 2021), which consists of satellite images and street-view panoramas collected from widely accessible map platforms. We do not gather or release any personally identifiable information (PII). All data sources follow the original dataset licenses and terms of use, and no effort is made to identify individuals or private

properties beyond what is already visible in the released benchmarks. Potential misuse of our approach should be considered: while our method advances urban-scale 3D scene reconstruction for beneficial applications such as autonomous driving simulation, AR/VR urban planning, and geographic visualization, it could also be applied to large-scale surveillance if deployed irresponsibly. To mitigate such risks, we emphasize that our framework is intended solely for academic research and positive societal use cases, and we release neither additional sensitive data nor pre-trained models tied to private regions.

## REPRODUCIBILITY STATEMENT

We aim to ensure the reproducibility of our results. All implementation is based on standard deep learning frameworks (PyTorch (Paszke et al., 2019)), and our model architecture, training strategies, and evaluation protocols are fully described in Sec. 4 and Appendix F. We specify dataset splits, preprocessing steps, and evaluation metrics in detail, and we adopt widely used benchmarks (VIGOR) to facilitate fair comparisons. Hyperparameters, batch sizes, number of iterations, and hardware configurations are reported in the Implementation Details paragraph. We will release the training scripts, configuration files, and inference code, together with instructions for data preparation and evaluation, upon publication. This enables other researchers to reproduce our quantitative scores and qualitative visualizations, and to extend our method to new geographic regions or related cross-view generation tasks.

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

This appendix provides more details and results of Sat3DGen.

## LLM USAGE

In this section, we clarify the role of large language models (LLMs) in preparing this work. The model was used exclusively for language polishing, such as refining grammar, style, and readability, without contributing to the research design, analysis, or conclusions.

## A  MORE 3D AND VIDEO RESULTS

### A.1  MESH RESULTS COMPARED TO SAT2DENSITY++

We provide additional mesh results in Fig. 6, with all satellite images sourced from the VIGOR-OOD test set.

### A.2  MESH RESULTS COMPARED TO CANONICAL IMAGE-TO-3D

As shown in Fig. 7, the Canonical Image-to-3D baseline produces significantly inferior geometry compared to our full model. Specifically, its ground surfaces are noisy and uneven, and it fails to capture distinct shapes for trees or produce flat building rooftops. The mesh boundaries also exhibit irregular, spiky extrusions. These geometric flaws result in a much lower rate of watertight meshes. In contrast, our model consistently generates smoother surfaces, more plausible object structures, and cleaner boundaries. This visual evidence strongly corroborates the quantitative results in Table 2, where the baseline's poor geometric fidelity is reflected in its high RMSE score.

### A.3  PANORAMA VIDEO RESULTS.

In the supplementary ZIP archive, the 'video_result' folder contains results from 16 sets of VIGOR-OOD test data. Each video is named according to its latitude and longitude, allowing you to view the latest satellite images by entering these coordinates into Google Maps[2]. In each video, the top-left corner displays the satellite image and camera trajectory, the top-right section presents the results of the Sat2Density++ (Qian et al., 2026), and the adjacent section showcases our results. From the presented videos, it is clear that our method, by producing enhanced 3D assets, achieves superior panorama video generation—yielding more regular building shapes, fewer floating artifacts, and improved generation of ground vehicles, road signs, and trees.

### A.4  SATELLITE-VIEW DEPTH COMPARISON

We present a qualitative comparison of the generated depth maps in Fig. 8. It is important to note the inherent challenge of temporal misalignment between the satellite images and the ground truth DSM, which was collected over a year (Section D). This leads to discrepancies from transient objects (e.g., vehicles) and noise within the DSM itself, making a perfect reconstruction unattainable.

Despite these challenges, a clear difference emerges. Sat2Density++ (b) yields overly-smoothed results with indistinct building edges and noisy ground planes. In contrast, our model (c) generates significantly sharper and more geometrically plausible structures, featuring flat rooftops and clean ground surfaces. These visual improvements directly corroborate our superior quantitative metrics reported in the main paper, demonstrating a better capability to reconstruct high-fidelity geometry from a single image.

## B  MORE APPLICATIONS RESULTS.

### B.1  SINGLE SATELLITE IMAGE TO DSM (METRIC DEPTH)

AS shown in Fig. 9, our model can render satellite-view metric depth from the learned NeRF-based 3D representation, even though no metric-depth annotations are used during training.

---

[2]https://www.google.com/maps/

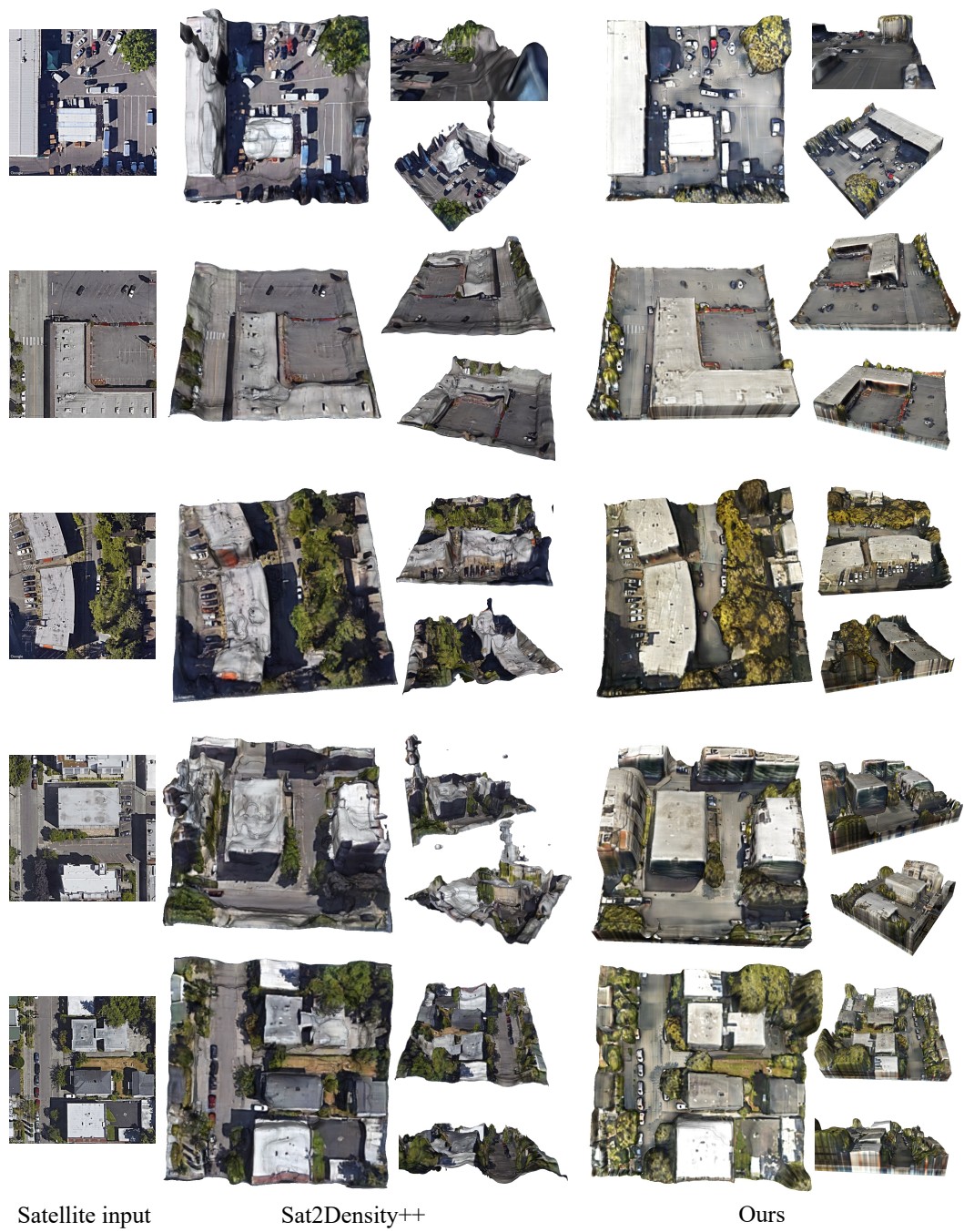

Satellite input    Sat2Density++       Ours

Figure 6: Comparison of generation 3D assets between Sat2Density++(Qian et al., 2026) and Ours.

### B.2   SURROUND-VIEW MULTI-CAMERA VIDEO GENERATION FROM A SINGLE SATELLITE IMAGE.

We provide some surround-view multi-camera video results in Fig. 4 and the supplementary ZIP archive, with four fixed 120-degree FOV perspectives shown at the bottom of each video. Our algorithm leverages NeRF-based 3D representations, allowing the generation of perspective images/videos with varying FOV and image sizes. The videos demonstrate that the quality of our

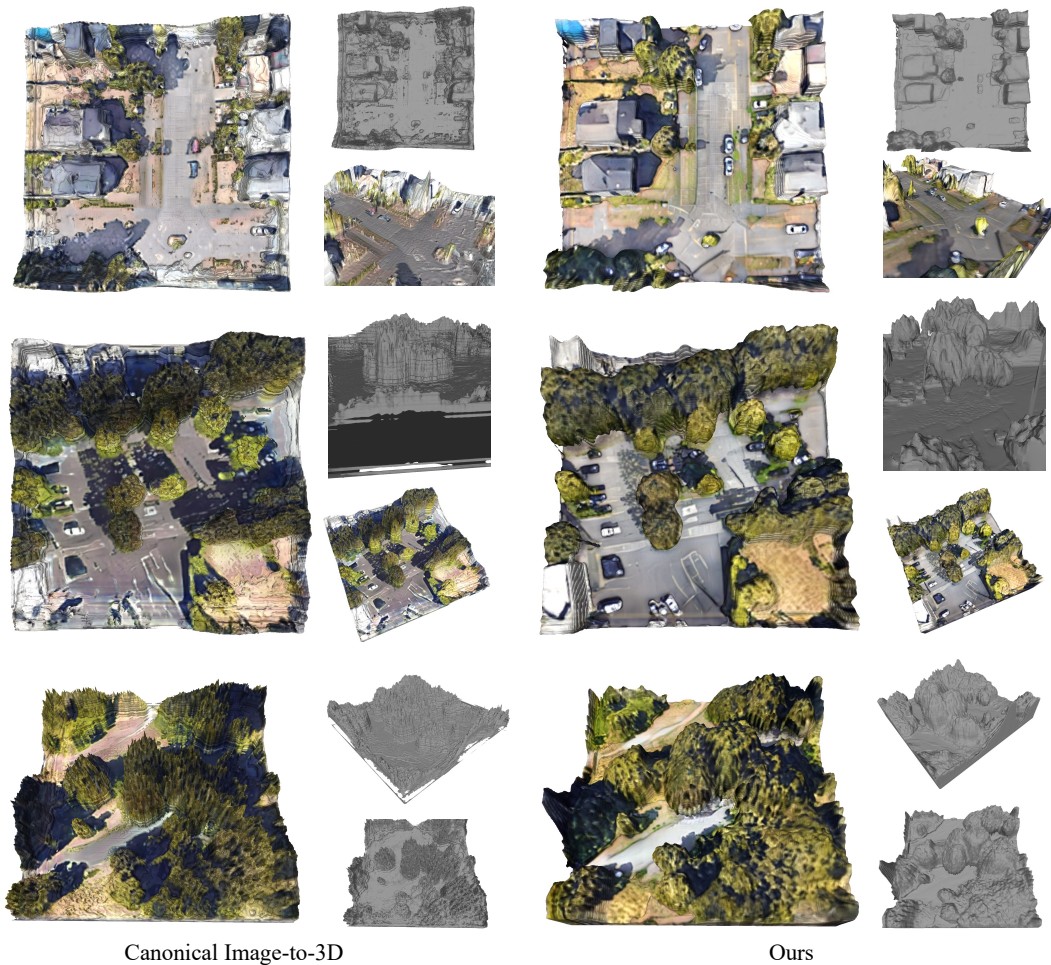

Canonical Image-to-3D                                          Ours

Figure 7: Comparison of generation 3D assets between the baseline Canonical image-to-3D model and Ours.

generated perspective images is nearly on par with panorama images, underscoring the superiority of our model design.

To the best of our knowledge, our approach is the first to generate diverse content in multi-view perspective videos from a single satellite image without requiring video data or 3D geometry as training input. The most relevant existing method for generating perspective videos from a single satellite image is Sat2Scene (Li et al., 2024b). However, this method is limited to synthesizing buildings and ground surfaces, primarily due to its strong reliance on building height maps converted into point clouds. Moreover, as their code and dataset are not fully open-sourced, a direct comparison with our results is not feasible. In summary, we demonstrate the capability of our algorithm to generate multi-view perspective videos, underscoring its potential applications in vehicle driving simulation.

### B.3 LARGE-SCALE MESH GENERATION

The process for generating a big mesh from a large satellite image is as follows: We first download an extensive satellite image from online platforms and then resize it to ensure the per-pixel resolution remains consistent with the pixel resolution of our training data. We then perform inference using a sliding window approach, processing each patch. Each patch resolution is 256x256 with a step set to 128. For the density at the edges, we simply average the values to obtain the final result. In our demo, as shown in Fig. 10, the input for the large-scale satellite image is at zoom level 19, covering a spatial

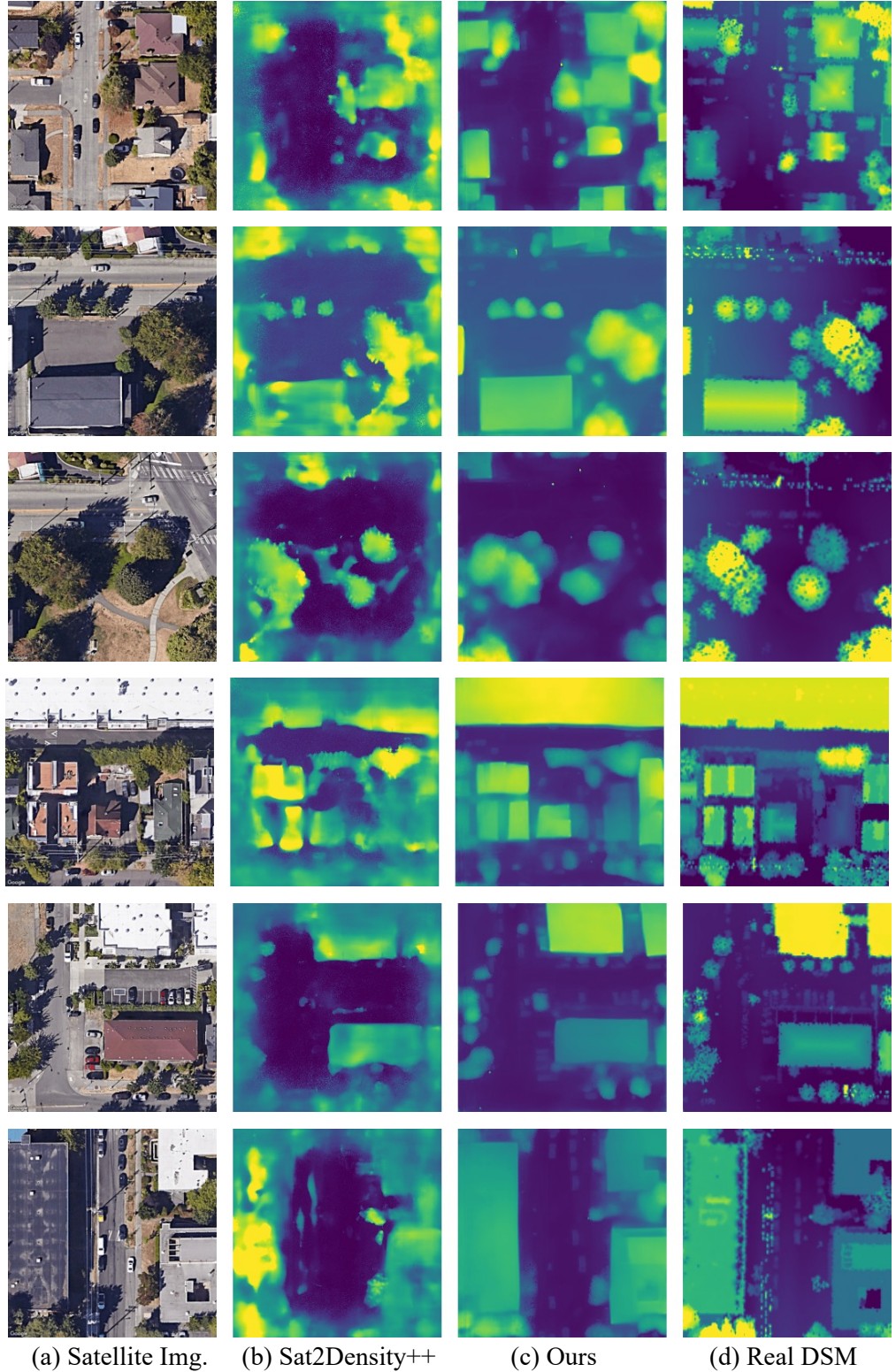

(a) Satellite Img.  (b) Sat2Density++  (c) Ours  (d) Real DSM

Figure 8: Comparison of predicted satellite-view height map.

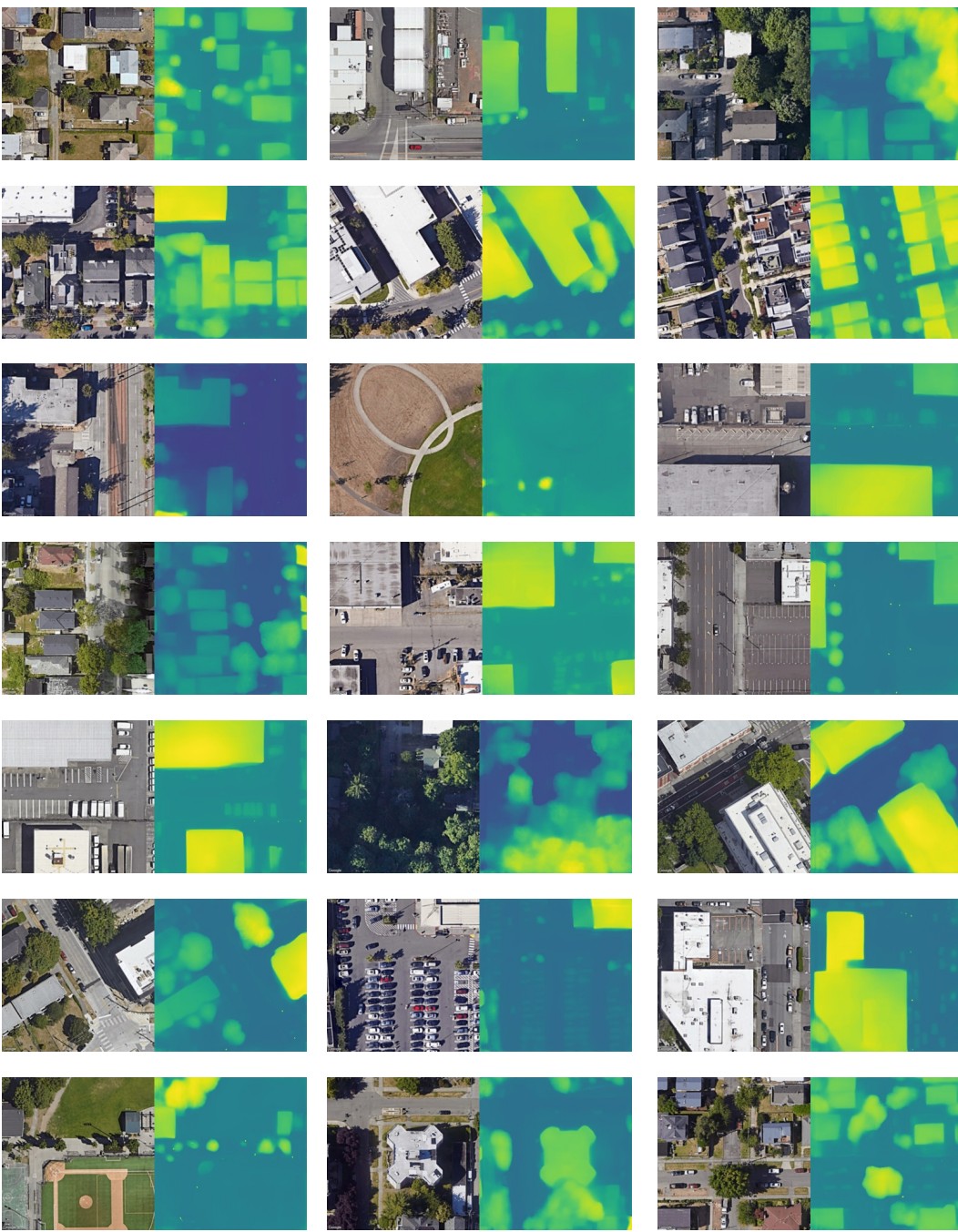

Figure 9: Visual results of our model generated DSM (metric depth) from the monocular satellite image, which is rendered from the satellite view, with no metric depth data for training.

area of approximately $150\,m \times 150\,m$. In theory, we can download larger remote sensing images to generate results over even broader areas. However, at a fixed zoom level, the Google Maps Static API returns at most 640×640 pixels per request.

Our demo results clearly demonstrate that the proposed algorithm produces smooth and coherent ground surfaces, seamlessly integrating multiple patches. The transitions between buildings show minimal discontinuities, highlighting the effectiveness of our method and its strong potential for large-scale 3D content generation.

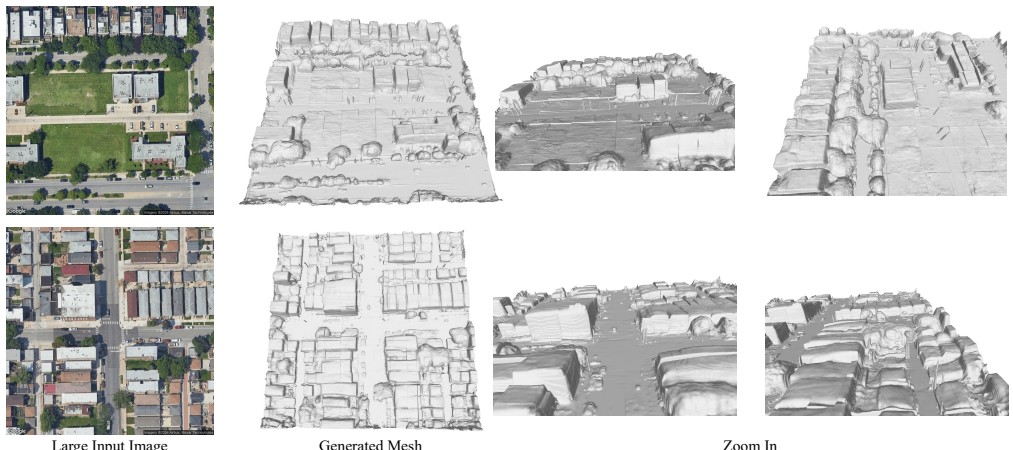

Large Input Image    Generated Mesh    Zoom In

Figure 10: Given a large satellite image, our model can generate mesh with sliding window inference mode.

### B.4 Semantic Maps to 3D Assets.

Generating 3D scenes from 2D semantic maps is a highly effective application. We can utilize open street map data to obtain ground semantic maps or directly create semantic maps through drawing. In our work, we collected multiple semantic map-satellite image pairs using OpenStreetMap[3] and Google Satellite Maps[4] to train an additional model for transforming color semantic maps into satellite images. This model is based on diffusion (Song et al., 2021), composed of ControlNet (Zhang et al., 2023) and SDXL (Podell et al., 2024).

As shown in Fig. 11, given a colored semantic map, the diffusion model first generates a satellite map, followed by Sat3DGen generating 3D assets based on the satellite image input. The results indicate that the generated 3D assets maintain spatial consistency with the semantic positions in the input semantic map. This application of generating 3D assets from semantic maps can be advantageous in spatial planning and game modeling, underscoring the significance of our work.

## C Implementation Details

**Perspective sampling.**   During training, we randomly obtain perspective images from panorama street view images. We set the pitch range to [-30, 30] and fix the roll at 0. The yaw is selected within the range [-79, 179], and the field of view (FOV) is randomly chosen from [90, 105, 120]; the render size is $256 \times 256$.

**Volume Rendering.**   The triplane dimension is set to 32, and there are 96 samples per ray.

**Other Hyperparameters.**   In our reconstruction of the panorama and satellite view perspectives, as well as in the sections of GAN loss and opacity loss, we maintain the same weight settings as Sat2Density++. For the newly introduced *Gravity-based Density Variation Loss*, we assign a weight of 3.5. As for the Satellite View Depth Regularization based on Midas loss, we set the weight to 0.1, acknowledging that the pseudo-labels for relative depth predicted by existing models may not be entirely accurate, hence opting for a smaller weight to mitigate potential adverse effects on the model.

Regarding the perspective view image reconstruction loss, we assign a weight that is half of the panorama view reconstruction loss. Conversely, the weight for the perspective view GAN loss is kept consistent with the panorama view GAN loss weight. This approach is due to the significant detail present in perspective images and the substantial difference between input satellite views and output perspective views. Given that the satellite input offers limited effective information for such

---

[3] https://www.openstreetmap.org/
[4] https://mapsplatform.google.com/

| Input semantic map | Synthesized sat. | Generated 3D by ours |
| --- | --- | --- |

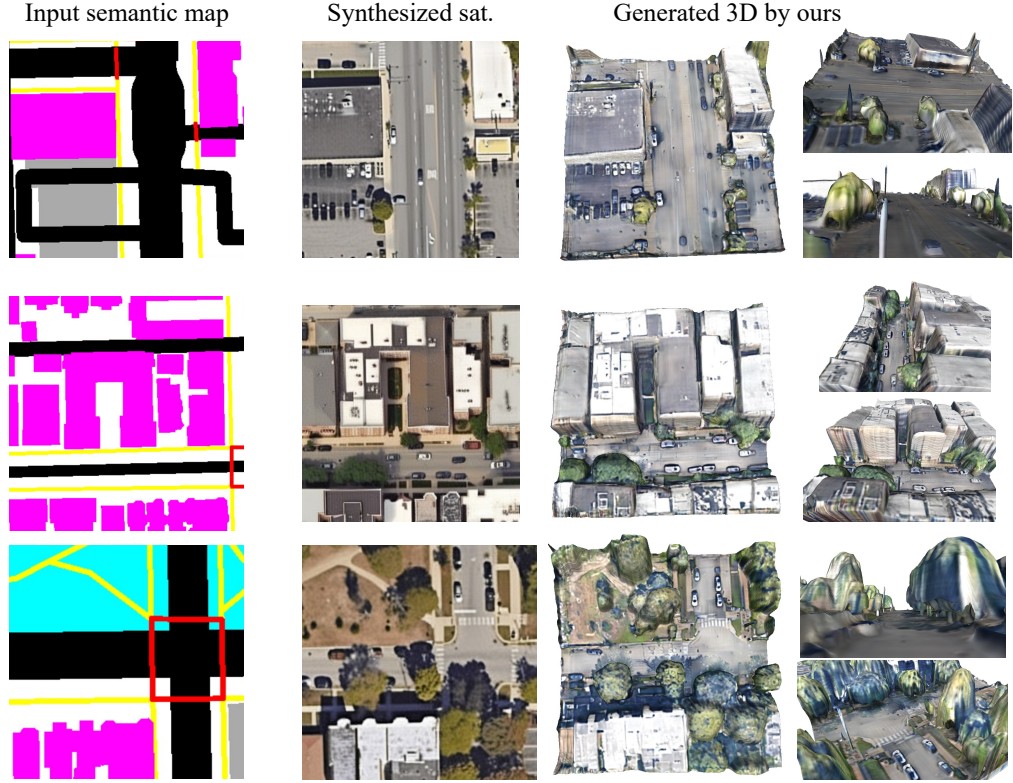

Figure 11: Given a colored semantic map, our model can generate 3D mesh through a pipeline that first converts the semantic map to a satellite map and then transforms the satellite map into 3D assets.

detailed perspective views, we aim to prioritize realism and image quality in generating perspective images (as constrained by GAN loss) rather than achieving visual consistency with the ground truth images (which is the target of the reconstruction loss constraint). The full code will be released after acceptance.

## D  GROUND TRUTH DSM PREPARATION FOR VIGOR-OOD

Quantitative evaluation of 3D geometry requires accurate ground truth Digital Surface Models (DSMs). However, high-quality, publicly available LiDAR-derived DSM data is scarce and typically limited to a few cities. We were fortunate that our out-of-distribution (OOD) test set, VIGOR-OOD, is based on the city of Seattle, for which we were able to obtain a corresponding high-precision DSM dataset. This section details the entire pipeline for processing and aligning this raw data to create the ground truth for our geometric evaluation.

### D.1  DATA SOURCE AND JUSTIFICATION

The ground truth DSMs were derived from the **King County West 2021** dataset, which is publicly available through the Washington State Department of Natural Resources (DNR) Lidar Portal[5]. As shown in the data's official report, this consists of Quality Level 1 (QL1) LiDAR data collected in the spring and summer of 2021. We specifically chose this data as its acquisition timeline closely matches the period when the VIGOR dataset (Zhu et al., 2021) was being created.

We selected six large GeoTIFF tiles from this collection that collectively cover the geographical extent of the VIGOR-OOD test images in Seattle, as visualized in Fig. 12. The official metadata

---

[5]https://lidarportal.dnr.wa.gov/

confirms the high quality of this source data, reporting positional errors of less than 5.6 cm with 95% confidence, which provides a reliable basis for our geometric evaluation.

## D.2 METHODOLOGY FOR SATELLITE-DSM ALIGNMENT

A significant technical challenge lies in aligning the raw DSM GeoTIFF files with the individual satellite images from the VIGOR-OOD test set. The datasets use different Coordinate Reference Systems (CRS), resolutions, and are not spatially aligned. We developed a robust processing pipeline to address this, which is outlined in Algorithm 1.

The key to our approach is leveraging the metadata provided by the VIGOR dataset, which, fortunately, includes the precise WGS84 latitude/longitude coordinates and the Google Maps zoom level for each satellite image. This allows us to first estimate the geographic bounding box of each image. We then reproject the corresponding section of the high-resolution DSM onto the exact pixel grid of the satellite image, ensuring perfect alignment. The main steps are detailed below.

---

**Algorithm 1** DSM Ground Truth Preparation Pipeline

---

1: **Input:** Set of VIGOR-OOD satellite images $I$, set of raw DSM GeoTIFF tiles $T$.
2: **Output:** A ground truth DSM array $D_i$ for each valid input image $I_i$.
3: Initialize an in-memory spatial index `dsm_index` for all tiles in $T$ for fast lookups.
4: **for** each satellite image $I_i$ in $I$ **do**
5:     {Step 1: Estimate satellite image's geographic footprint}
6:     Parse center latitude/longitude and zoom level from $I_i$'s filename.
7:     Estimate the WGS84 bounding box `target_bounds` for $I_i$.
8:     {Step 2: Find and Reproject Overlapping DSM Data}
9:     Find all candidate tiles `candidate_tiles` from `dsm_index` that spatially overlap with `target_bounds`.
10:     **if** `candidate_tiles` is empty **then**
11:       **continue** {No DSM coverage for this image}
12:     **end if**
13:     Initialize `best_dsm` to 'null' and `max_coverage` to 0.
14:     **for** each tile $T_j$ in `candidate_tiles` **do**
15:       Create an empty destination array `temp_dsm` with the same dimensions as $I_i$.
16:       Reproject the data from $T_j$ onto `temp_dsm` using bilinear resampling.
17:       **if** coverage of `temp_dsm` > `max_coverage` **then**
18:         `best_dsm` ← `temp_dsm`, `max_coverage` ← coverage.
19:       **end if**
20:     **end for**
21:     {Step 3: Post-processing and Quality Control}
22:     **if** `best_dsm` is not 'null' **then**
23:       Convert elevation values in `best_dsm` from feet to meters.
24:       Calculate the percentage of NaN pixels `nan_percent` in `best_dsm`.
25:       **if** `nan_percent` ≤ 5.0% **then**
26:         Save `best_dsm` as the final ground truth $D_i$.
27:       **else**
28:         Discard this pair due to insufficient DSM coverage.
29:       **end if**
30:     **end if**
31: **end for**

---

This automated pipeline ensures that for every test image, we generate an aligned, unit-corrected, and quality-controlled DSM that can be used for direct, pixel-wise comparison in our geometric accuracy evaluation. Fig. 8 shows visualizations of several paired satellite images and their corresponding DSM data.

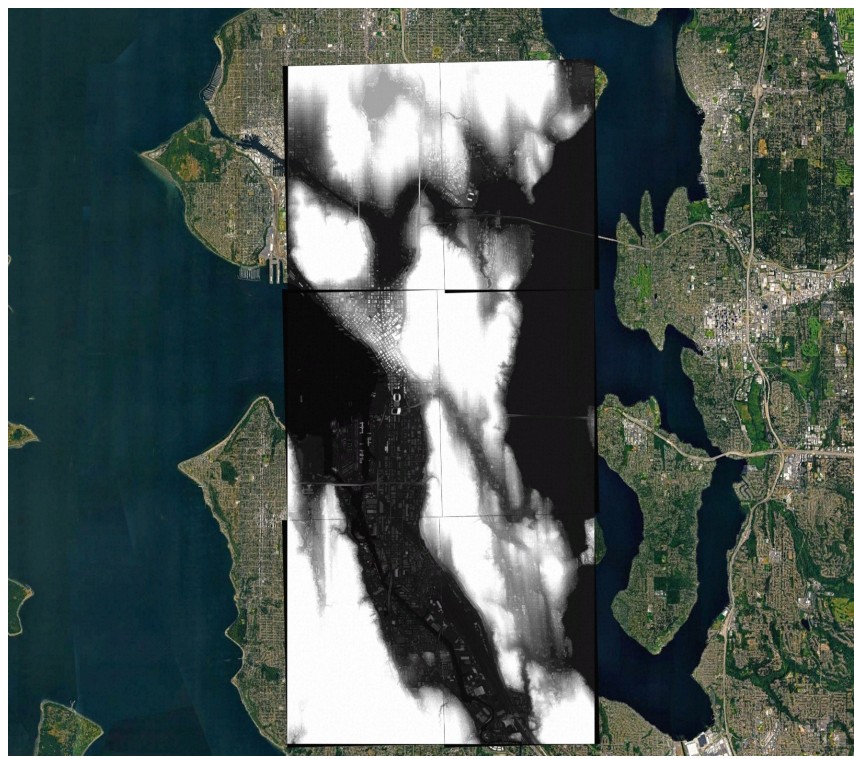

Figure 12: The collected DSM data in Seattle City.

Table 4: Ablation on variation regularization. "No variation loss" means starting from the base (Full model w/o perspective training) and removing $\mathcal{L}_{\text{grav}}$. "w. TV loss" replaces $\mathcal{L}_{\text{grav}}$ with total variation (TV) loss Chan et al. (2022). The remaining rows use $\mathcal{L}_{\text{grav}}$ with different $\epsilon$; our choice $\epsilon$=1.0 (Ours) performs best overall. Lower is better.

| Method | FID↓ | KID×100↓ |
|---|---|---|
| No variation loss | 25.90 | 19.0 |
| w. TV loss | 24.83 | 20.1 |
| $\mathcal{L}_{\text{grav}}$ ($\epsilon$=0) | 24.52 | 18.7 |
| $\mathcal{L}_{\text{grav}}$ ($\epsilon$=0.01) | 22.61 | 17.2 |
| $\mathcal{L}_{\text{grav}}$ ($\epsilon$=0.1) | 22.63 | **16.2** |
| $\mathcal{L}_{\text{grav}}$ ($\epsilon$=0.5) | 21.94 | 17.9 |
| $\mathcal{L}_{\text{grav}}$ ($\epsilon$=1.0) | **21.60** | **16.2** |
| $\mathcal{L}_{\text{grav}}$ ($\epsilon$=5.0) | 21.74 | 18.5 |
| $\mathcal{L}_{\text{grav}}$ ($\epsilon$=10.0) | 21.66 | 17.5 |

# E    MORE ABLATIONS

## E.1    VARIATION REGULARIZATION ABLATION.

As shown in Table 4, removing variation regularization gives the worst results (FID 25.90, KID 19.0). Replacing it with a TV loss slightly lowers FID (24.83) but hurts KID (20.1), indicating oversmoothing and weak structural guidance. Our $\mathcal{L}_{\text{grav}}$ outperforms both, with $\epsilon = 1.0$ yielding the best metrics (FID 21.60, KID 16.2). Setting $\epsilon = 0$ degrades performance because it fails to accommodate genuine voids (e.g., gaps beneath tree canopies outside the trunk), leading to over-penalization and smoothing.

## F    DATASET PREPARATION AND DETAILS

**VIGOR.** VIGOR (Zhu et al., 2021) contains four cities and, for each satellite image, multiple associated street-view panoramas with known relative poses. The satellite zoom level is 20. We train in Chicago, New York, and San Francisco, and use Seattle as an unseen-city OOD test set. We use 78,188 satellite–panorama pairs for training and 11,875 pairs for metric evaluation. In principle, additional cities could be incorporated to further scale training.

**Discussion on other Datasets.** CVACT (Shi et al., 2022) and CVUSA (Zhai et al., 2017) are widely used for satellite-to-street image synthesis and localization. However, they are unsuitable for our 3D generation task due to two main reasons.

First, their data structure provides **insufficient geometric supervision**. Each satellite tile in these datasets is paired with a single ground-level image at the tile center, without multi-view coverage. This data design is incompatible with our method, which relies on panoramas captured at spatially distinct positions to learn consistent street-level 3D. With only a centered street view, the supervision is too sparse to learn reliable geometry, rendering our optimization components ineffective. Consequently, we do not use CVACT or CVUSA for training. In practice, VIGOR-style datasets can be constructed for any geographic region where street-level panoramic imagery is available (e.g., via Google Maps) by pairing a single satellite image with N panoramic views.

Second, these datasets are unsuitable for a fair out-of-distribution (OOD) evaluation. The satellite images in CVACT and CVUSA were captured by different satellites and at different zoom levels than those in VIGOR. This results in significant appearance shifts and differences in pixel-level spatial resolution. Evaluating our VIGOR-trained model on these datasets would conflate the test of geometric generalization with a test of resilience to data source domain shift, which is beyond the scope of this work. We test on an unseen city from the same data source.

**Our Strategy for Generalization Evaluation.** We explicitly test generalization using the VIGOR-OOD split, by training on three cities and testing on the unseen city of Seattle. This setup introduces a significant domain gap in terms of urban layouts and architectural styles, providing a strong signal of our model's robustness. Furthermore, the notion that VIGOR is purely "urban" is a misconception. Like other datasets derived from vehicle-based captures, it covers a wide range of environments, including many less-dense, suburban areas, as shown in our qualitative results (e.g., Figure 7). The primary generalization challenge is not urban vs. suburban density, but rather the architectural and environmental shifts between disparate geographic regions.

## G    LIMITATIONS

Our work faces challenges rooted in both data availability and model assumptions.

**Pose Inaccuracy.** A primary limitation is the lack of precise pose data. We treat satellite images as ideal orthogonal projections, which is not always the case. Moreover, the panoramas lack authentic intrinsic/extrinsic parameters; we only have GPS data. Our model assumes panoramas are captured perpendicular to the ground, ignoring roll angles from terrain or road banking. Accessing or predicting accurate poses is a valuable direction for future work.

**Geometric and Terrain Assumptions.** Our model's performance is also constrained by its underlying assumptions. By its generative nature, it struggles with **atypical architectures** that are rare in the training data, as we lack explicit 3D ground-truth shapes for supervision. Furthermore, our framework assumes a **locally flat ground plane** and does not model significant terrain variations like hills, which are difficult to infer from sparse imagery alone. Future work could address this by incorporating multi-modal data, such as terrain maps.

**Evaluation Metrics.** Finally, while metrics like multi-view photometric consistency or temporal flicker are powerful, they are not applicable to the VIGOR dataset. VIGOR's sparse, non-sequential collection of still images does not support the evaluation of temporal stability or dense view consistency.

