# OpenReview forum: "Sat3DGen: Comprehensive Street-Level 3D Scene Generation from Single Satellite Image"
_ICLR.cc/2026/Conference — ICLR 2026 Poster_

### Official Review · Reviewer_tZHZ · 2025-10-21

**Soundness:** 2
**Presentation:** 1
**Contribution:** 2
**Rating:** 2
**Confidence:** 5

**Summary:**

The paper introduces Sat3DGen, a framework designed to generate comprehensive street-level 3D scenes from a single satellite image. The method incorporates several techniques — a gravity-based density variation loss, spatial token padding, and a monocular relative-depth prior — to enhance the performance of Sat2Density++. However, the overall architecture remains almost identical to Sat2Density, leading to concerns about the lack of novelty.

**Strengths:**

1. The authors tackle a meaningful and challenging task: generating realistic ground-level 3D scenes from a single satellite image.
2. The proposed gravity-based density variation loss, spatial token padding, and monocular relative-depth prior improve upon the previous Sat2Density++ framework.

**Weaknesses:**

**1. The use of DINO-v3**

The model employs DINO-v3, which is computationally expensive. It is unclear how the inference speed and GPU memory consumption are affected. Moreover, it remains uncertain whether the performance gains primarily come from DINO-v3, rather than the proposed method itself.

**2. Unclear explanation of the Gravity-based Density Variation Loss (Lines 241–254)**

* The mathematical formulation is ambiguous. Is x defined in Line 197 as a 3D point?
* Why does δx (a scaled 3D point) represent “along gravity”?
* The statement “lower-altitude points usually have density that is no smaller than higher-altitude points” seems empirical — is there any theoretical justification?

**3. Depth estimation and supervision issues (Line 258–265)**
* How accurate is the depth estimated by Depth Anything v2 when applied to satellite imagery?
* What happens if the estimated depth is inaccurate?
* Why is depth only used as a loss term instead of being fused into the network representation?
* The supervision on spatial gradients (Line 265) might oversmooth regions that should exhibit sharp depth changes (e.g., building-ground boundaries).

**4. Inconsistent baseline comparison**

* In Figure 4(c), the comparison is made against Sat2Density instead of Sat2Density++. Why not compare with the most recent and stronger baseline?
* The paper lacks quantitative comparisons with related methods such as ControlNet, ControlS2S, or Canonical Image-to-3D.

**5. Limited novelty**

The overall architecture is nearly identical to Sat2Density++, with improvements mainly stemming from new loss terms and a stronger backbone.

**Questions:**

The problem corresponds one-to-one with the content of the weakness:
1. How does the use of DINO-v3 affect inference speed and memory consumption? Are the improvements mainly due to the stronger feature extractor?
2. Could the authors provide clearer mathematical formulations and theoretical justification for the Gravity-based Density Variation Loss?
3. How accurate is Depth Anything v2 on satellite imagery, and how sensitive is the model to potential depth estimation errors? Why not integrate depth directly into the model rather than using it as a loss? Does the spatial gradient supervision risk over-smoothing sharp depth transitions?
4. Why is Sat2Density used as the baseline in Figure 4(c) instead of Sat2Density++? Why are quantitative comparisons with ControlNet, ControlS2S, or Canonical Image-to-3D missing?
5. How do the authors justify the novelty of Sat3DGen given its high similarity to Sat2Density++?

**Details Of Ethics Concerns:**

Non-existent

---

> ### Author Response · Authors · 2025-11-22
> **Responses to Reviewer tZHZ (1/3)**
>
> ## Q1/W1: The use of DINO-v3:
>
> **Response: On the Impact and Attribution of DINO-v3**
>
> We thank the reviewer for these insightful questions. We address them directly below.
>
> **1. Impact of DINO-v3 on Inference Speed and Memory**
>
> Our choice to use a frozen DINO-v3 backbone is a strategic one, following the highly successful practice in recent state-of-the-art  object level image-to-3D generation. This approach leverages a powerful pre-trained model for a robust feature foundation.
>
> To quantify the trade-offs, we conducted a targeted ablation study:
>
> | Encoder | Trainable Params | Encoder Speed (FPS)↓ | $L _{dep}$ | $L _{grav}$ | Sp. Tok. | Per. Train | FID↓ |
> | --- | --- | --- | --- | --- | --- | --- | --- |
> | VAE-style (Trainable) | ~156M | 33 | ✓ | ✓ | ✓ | ✓ | 29.0 |
> | DINOv3-Base (Frozen) | ~76M | 30 | ✓ | ✓ | ✓ | ✓ | 23.0 |
> | **DINOv3-Large (Frozen)** | ~76M | 23 | ✓ | ✓ | ✓ | ✓ | **19.2** |
>
> The data confirms our strategy is effective:
>
> *   While using the larger, more powerful backbone results in a modest decrease in inference speed, it yields a **crucial and dramatic improvement in generation quality** (FID dropping from 29.0 to 19.2).
>
> *   We believe this is a well-justified and necessary trade-off for achieving the high-fidelity results our method delivers.
>
>
> **2. The Source of Performance Gains: Our Methodology**
>
> This is the key point: **the primary performance gains are driven by our proposed methodology, not just the stronger backbone.**
>
> Our main ablation study (revised Table 2) isolates this effect perfectly. In that study, we use the **same DINOv3 backbone for all experiments**, including the baseline.
>
> *   The baseline (with DINOv3) scores an FID of **35.6**.
>
> *   Our full model (also with DINOv3) improves the FID to **19.2**.
>
>
> | Model | DINOv3 | $L _{dep}$ | $L _{grav}$ | Sp. Tok. | Per. Train | FID↓ | KID×100↓ | RMSE↓ |
> | --- | --- | --- | --- | --- | --- | --- | --- | --- |
> | Canonical Image-to-3D | ✓ |  |  |  |  | 35.6 | 30.1 | 6.21 |
> | Base w/o $L _{dep}$ | ✓ |  | ✓ | ✓ |  | 23.7 | 18.4 | 5.75 |
> | Base w/o $L _{grav}$ | ✓ | ✓ |  | ✓ |  | 25.9 | 19.0 | 5.21 |
> | Base w/o Spatial Tokens | ✓ | ✓ | ✓ |  |  | 24.8 | 18.1 | 5.64 |
> | Base (Full w/o per. training) | ✓ | ✓ | ✓ | ✓ |  | 21.6 | 16.2 | 5.23 |
> | Full model | ✓ | ✓ | ✓ | ✓ | ✓ | 19.2 | 13.6 | 5.20 |
>
> This ~16-point FID improvement comes entirely from our novel geometric constraints and training strategies. As detailed in our general response (under **"Clarification of Novelty..."**), this holistic methodology is our core contribution.
>
> In essence, DINO-v3 provides the strong **foundation**, but our **methodology** is the architect that builds the high-quality 3D scene and is the primary source of our breakthrough results.
>
> ---
>
> ## Q2/W2: Clarity on the Gravity-based Density Variation Loss**
>
> Thank you for your precise feedback, which helped us significantly clarify the motivation and formulation of our Gravity-based Density Variation Loss. We have revised the paper to address each of your questions directly:
>
> *   **"Is x defined as a 3D point?"**
>
>     Yes. We now explicitly state that $\mathbf{x}$ is a 3D point in $\mathbb{R}^3$.
>
> *   **"Why does δx represent 'along gravity'?"**
>
>     This was an ambiguity in our original text. We have now replaced the generic $\delta\mathbf{x}$ with the more precise $\delta\mathbf{z}$ to denote a small upward displacement vector purely along the z-axis (i.e., anti-gravity). The loss is named for the physical principle it is inspired by, not the vector's direction.
>
> *   **"Is the density assumption empirical or theoretical?"**
>
>     Our regularizer is motivated by the **empirical observation** that plausible 3D scenes are generally denser at the bottom (e.g., contrasting dense **solid ground and tree trunks** with sparser **leafy canopies**). We believe this common pattern is not coincidental but has a clear **physical basis**.
>
>     As we now clarify in the paper, the reasoning is twofold: 1) In predominantly opaque outdoor scenes, NeRF's volume density ($\sigma$), which measures light obstruction, serves as a natural proxy for physical matter. 2) The principle of gravity dictates that this matter tends to accumulate at lower elevations.
>
>     Therefore, by grounding the empirical observation in this physical rationale, our loss functions as a **principled regularizer**, rather than a simple heuristic.

---

> ### Author Response · Authors · 2025-11-22
> **Responses to Reviewer tZHZ (2/3)**
>
> ## Q3/W3: Depth estimation and supervision issues
>
> We thank the reviewer for these insightful questions, which allow us to clarify the motivation and function of our depth regularization.
>
> **1. On the Accuracy and Nature of Depth Supervision (1. How accurate is the depth estimated by Depth Anything v2 when applied to satellite imagery? 3. Why is depth only used as a loss term instead of being fused into the network representation?):**
>
> Depth Anything v2 (DAv2) produces **relative depth**, which captures the ordinal relationships between pixels (i.e., pixel A is farther than pixel B) but lacks a consistent metric scale. In contrast, our model renders **metric depth** within its 3D scene representation.
>
> We reports the performace of DAv2 here:
>
> | MAE↓ | RMSE↓ | < 2.5m ↑ | < 7.5m ↑ |
> | --- | --- | --- | --- |
> | 3.12 | 4.37 | 65.62% | 91.90% |
>
> The reported performance is obtained after scale-and-shift alignment with the ground truth, and therefore is not directly comparable to metric depth prediction methods (including ours). Our loss is designed to capitalize on this structural signal, not its non-metric values.
>
> This fundamental difference is precisely **why we chose a scale-and-shift invariant loss (MiDaS loss)** and **why we use it as a loss term instead of fusing it into the network**. Fusing a relative depth map directly into a metric 3D representation is an ill-posed problem, as it would require learning a complex, scene-dependent transformation. Using it as a loss, however, provides a more robust and direct way to inject DAv2's valuable structural prior without forcing this difficult alignment.
>
> **2. What happens if the estimated depth is inaccurate?**
>
> Thank you for this excellent question. The impact of inaccurate depth is minimal because it acts as a **low-weight regularizer** in our model.
>
> Our primary training objective is to generate a 3D scene that is photometrically consistent with **both the satellite and street-view images**. However, inferring a complete 3D structure from two such sparse and disparate views is an extremely ill-posed problem. The depth term provides a crucial geometric prior to guide the optimization and resolve ambiguity, ensuring a plausible 3D shape.
>
> If the estimated depth is inaccurate, it will likely conflict with the strong visual evidence from at least one of the input images. Since the photometric losses are the dominant objective, the model will prioritize satisfying them. In short, it will learn to **ignore the incorrect depth guidance** in favor of rendering a scene that accurately matches the ground-truth images.
>
> **3. On the Gradient Term and Oversmoothing (Q4):**
>
> We thank the reviewer for this keen observation, as it touches upon a critical aspect of our chosen loss function. However, we respectfully believe there is a misunderstanding of how the gradient-matching term in Eq. 4 functions.
>
> The term `||∇(sD̂+t) - ∇D*||` **does not act as a general smoothing filter**. Instead, it encourages the gradient of our rendered depth (`∇D̂`) to **match** the gradient of the pseudo-relative-depth (`∇D*`).
>
> *   Therefore, where the pseudo-depth exhibits **sharp changes** (precisely at building-ground boundaries), this loss term **explicitly encourages our model to also produce sharp depth changes**.
>
> *   Conversely, only on surfaces that are smooth in the pseudo-depth (like flat rooftops or roads), does it penalize spurious bumps and irregularities in our output.
>
>
> In essence, the gradient term functions as a **structure-preserving regularizer**. It helps us replicate the sharp structural boundaries present in the DAv2 output, which is the exact opposite of oversmoothing them. This makes it an ideal choice for our task.

---

> ### Author Response · Authors · 2025-11-22
> **Responses to Reviewer tZHZ (3/3)**
>
> ## Q4/W4: Inconsistent baseline comparison
>
> **1. Correction of Inconsistent Baseline in Figure 4(c)**
>
> We apologize for the confusing typo in the caption of Figure 4(c).
>
> The reviewer is absolutely right that the comparison should be against the stronger **Sat2Density++** baseline. This was indeed our intention and what we implemented, as stated in the "Qualitative Comparison" paragraph of our original submission: _"We compared the panorama video results generated by Sat2Density++ and our model."_
>
> Regrettably, the caption's typo contradicted our own text and caused confusion. We have now corrected the caption in the revised manuscript.
> We appreciate reviewer's careful reading.
>
> **2. Quantitative Comparisons with Other Related Methods**
>
> We acknowledge the request for broader quantitative comparisons against methods like ControlNet, ControlS2S, and Canonical Image-to-3D. Here is our explanation for the comparison strategy:
>
> *  **ControlNet and ControlS2S**: A direct quantitative comparison is unfortunately infeasible for two main reasons:  **Lack of Public Code**: Neither ControlNet (for this task) nor ControlS2S （both score reported from ControlS2S paper\[1\]）have been open-sourced. Reproducing these diffusion-based models without the original code and training setup would be extremely challenging and likely result in an unfair, biased comparison.
>
>
> \[1\] Controllable satellite-to-street-view synthesis with precise pose alignment and zero-shot environmental control. ICLR, 2025.
>
> *  **Canonical Image-to-3D**: This model was designed as an internal, controlled baseline primarily for our ablation study (Table 1) to isolate the effects of our proposed components. Following the reviewers' valuable suggestion, we have now included additional **qualitative comparisons** against "Canonical Image-to-3D" in the supplementary material (Figure 7) to provide a more complete picture.
>
> ---
>
> ## W5/Q5: Novelty of Sat3DGen in Relation to Sat2Density++
>
> Thank you for this critical question regarding novelty.
>
> While our model shares a high-level architectural paradigm with Sat2Density++, we argue that our novelty is substantial and lies not in reinventing the overall pipeline, but in our **holistic, geometry-first methodology** designed to solve fundamental, previously unaddressed challenges in this task.
>
> Our key innovations are centered on a **synergistic suite of solutions**, including novel geometric losses ($L _{grav}$, $L _{dep}$), Spatial Tokens, and a new supervision strategy (Perspective View Training). Each component is meticulously designed to tackle specific failure modes like the extreme viewpoint gap and sparse supervision, which are inherent to satellite-to-street 3D generation.
>
> Crucially, the significant performance gains are **not merely an artifact of a stronger backbone**. Our comprehensive ablation study (Table 2) unequivocally demonstrates that removing our proposed components leads to a drastic drop in performance (e.g., FID increases from 19.2 to 23.7, 25.9, etc.), proving their individual and collective necessity.
>
> To make these distinctions crystal clear, we have prepared a detailed clarification, including a table. We respectfully invite you to review our **Clarification of Novelty and Detailed Comparison with Sat2Density++ in ''Summary of Responses to Reviewers (3-4/4)''** for a full discussion.

---

> > ### Comment · Reviewer_tZHZ · 2025-11-26
> >
> > Thank the authors for the detailed clarifications. I am satisfied with the authors’ responses to Q1, Q2, and Q3, and I encourage them to incorporate these clarifications into the paper.
> >
> > Regarding Q4, the authors state that ControlS2S is not open-sourced; however, the code was actually released five months ago. I understand that reproducing the ControlS2S results may not be feasible during the rebuttal period, but I hope the authors will include the reproduced results in the next version of the paper.
> >
> > Finally, the most critical issue lies in Q5. I still believe that the contributions of this paper are insufficient. I have also noticed that reviewers zCTK and udS7 express similar concerns, and I am willing to hear their perspectives.

---

> ### Author Response · Authors · 2025-11-26
>
> Dear Reviewer tZHZ,
>
> Thank you for your continued engagement and for providing further feedback on our work. We appreciate this opportunity to clarify the remaining points to ensure our contributions are accurately understood by yourself, the other reviewers, and the AC.
>
> ### On the ControlS2S Comparison
>
> We share your perspective on the importance of comparing with relevant state-of-the-art work. However, we face a significant practical barrier regarding ControlS2S, which we must bring to your attention.
>
> After a thorough search, we have been unable to locate any officially released code for ControlS2S. The paper's primary pages (both the [arXiv version](https://arxiv.org/pdf/2502.03498) and the [OpenReview page](https://openreview.net/forum?id=f92M45YRfh)) do not contain a official code link.
>
> You mentioned that "the code was actually released five months ago." To move forward, we kindly request your assistance in this matter. Could you please provide the specific link and search methodology that led you to this conclusion? Without access to a verified, official implementation, we cannot perform a scientifically sound comparison.
>
> Furthermore, we wish to reiterate a crucial point about the scope of our work versus ControlS2S. Our method is designed for comprehensive 3D-aware generation, producing coherent 3D geometry, continuous videos, and street-view images. In contrast, ControlS2S focuses on generating discrete, 2D street-view images. Its technical scope is a subset of ours, and it does not address the core challenges of 3D consistency and video synthesis that our paper tackles. We believe this distinction is fundamental when assessing the contributions of our work.
>
> ### On the Novelty of Our Approach
>
> Regarding the remaining concerns about novelty, we would appreciate specific feedback on which part of our rebuttal's argument you found insufficient, as this would enable a more targeted and productive discussion.
>
> To be clear, our core novelty lies in proposing a **geometry-first paradigm** for the satellite-to-3D task. This is a direct response to the geometric artifacts prevalent in existing feedforward image-to-3D methods. Our contributions are not just a collection of losses, but a **cohesive strategy** where each component synergistically works to enhance 3D geometry, which is **not only crucial for enabling a wide range of downstream applications but also directly improves 2D rendering quality**. We argue that novel training strategies and supervision schemes are as vital to advancing the field as novel architectures, and our work stands as a testament to this principle.
>
> ### On the Technical Assessment and Review Confidence
>
> Finally, we feel compelled to address a point of confusion that arose from the review process, which is critical for the overall evaluation of our paper.
>
> 1.  In the initial review (Q3/W3), our gradient-matching loss was flagged for potentially causing **oversmoothing**.
>
> 2.  In our rebuttal, we provided a detailed, technical explanation clarifying that this loss term's function is the **exact opposite** of smoothing. As is standard in 3D vision, its purpose is to force the **gradient field** of our predicted depth map to match the **structural gradients** of the pseudo-depth prior. In doing so, it explicitly encourages the replication of sharp geometric features (like building edges), rather than erasing them.
>
> 3.  In your latest response, this core technical point was not engaged with; you simply stated you were "satisfied" and moved on.
>
> This sequence is concerning. For a review with the highest confidence score (5/5), we would have anticipated a substantive dialogue on such a fundamental technical aspect, especially after a **clear misunderstanding was identified**. The lack of engagement on this point leaves us, and likely others following this discussion, wondering about the basis for the high confidence in the technical assessment.
>
> We are not raising this to be adversarial, but to ensure a fair and technically grounded evaluation. The discrepancy between the expressed confidence and the depth of engagement on a key technical pillar of our work is a factual point that we believe is relevant for the final deliberation.
>
> Thank you for your consideration. We remain committed to improving our paper and hope for a resolution based on a shared and accurate technical understanding.

---

> > ### Comment · Reviewer_tZHZ · 2025-11-27
> >
> > Regarding Q4, I have already sent the code link to the AC, and it is up to the AC to decide whether to forward the link to the authors.
> >
> > Regarding Q5, I notice that not only I but also reviewers zCTK and udS7 find that the basic framework of this paper is quite similar to Sat2Density++. I am willing to hear their perspectives on whether a work with insufficiently rigorous experiments and a framework that is quite similar to prior papers can meet the acceptance standards of ICLR.

---

> ### Author Response · Authors · 2025-11-27
>
> Dear Reviewer tZHZ,
>
> Thank you for your response and for clarifying the situation with the code link.
>
> **1. Regarding the ControlS2S Code (Q4):**
>
> To clarify, our initial question was purely a practical one, as we were trying to understand how to find the public code. We believe easy access to the code for important work like ControlS2S is highly beneficial for the community.
>
> We understand the link has been sent to the AC. We simply wished to highlight the accessibility point. Should our paper be accepted and we gain access to the code, we commit to including a comparison in the camera-ready version.
>
> **2. Regarding Experimental Rigor and Novelty (Q5):**
>
> Thank you for sharing your perspective on experimental rigor. We were very encouraged that the initial reviews consistently highlighted our "comprehensive validation" and "effective ablation studies" as a strong point of the paper. Building upon this strong foundation, the discussion process has further strengthened the paper, for instance, through the construction of a new quantitative benchmark.
>
> On the similarity to Sat2Density++, we have further articulated our core contribution as a **holistic, geometry-first methodology**. We believe our work follows a well-established path to novelty in machine learning and computer vision. Many impactful papers demonstrate their contribution not through entirely new architectures, but through targeted methodological innovations that solve critical flaws in prior work, such as new paradigms like **Diffusion Models** or influential objective functions like the **Perceptual Loss**. Ultimately, we believe the essence of scientific contribution lies in identifying a critical problem and then proposing an effective, targeted solution.
>
> While we have presented our case based on this view, we understand that different perspectives can exist, and we respectfully acknowledge yours.
>
> We hope our clarifications have been helpful and remain ready to address any further specific points you may have.

---

### Official Review · Reviewer_Quvg · 2025-10-31

**Soundness:** 3
**Presentation:** 3
**Contribution:** 3
**Rating:** 6
**Confidence:** 4

**Summary:**

This paper proposes Sat3DGen, a novel framework for generating high-fidelity, street-level 3D reconstructions from a single satellite image. The method introduces three geometry-focused contributions: (1) Gravity-based density variation loss, (2) Spatial token padding, (3) Monocular satellite-view depth regularization. It demonstrates substantial improvements over prior state-of-the-art approaches like Sat2Density++ and Sat2Scene, particularly in scene-level 3D geometry consistency, semantic fidelity, and rendered view realism, across both qualitative and quantitative benchmarks (e.g., FID, DINO similarity).

**Strengths:**

- Clearly motivated and well-structured paper with substantial methodological contributions.
- The paper achieves strong improvements in empirical results (e.g., FID drops from 40.8 to 19.2) and demonstrates practical utility in applications such as DSM estimation and multi-view video synthesis.
- Effective ablation studies and transparent discussion of limitations are provided.

**Weaknesses:**

- The paper lacks evaluation against metric 3D ground truth (e.g., DSM or city-scale LiDAR) and could be strengthened with controlled experiments on public datasets and analysis of DSM or mesh error.
- There is insufficient discussion of major failure modes (such as occlusions or challenging geometry), and additional metrics beyond FID/LPIPS—like multi-view photometric consistency or temporal flicker—would offer a fuller assessment of 3D and video realism.
- Robustness and generalization should be further explored, including evaluation on non-VIGOR data, handling of noisy or missing illumination inputs, and clarifications on methodological details and reproducibility (e.g., pretrained model release, citation updates, and clear diagram separation).

**Questions:**

1. Are the generated meshes watertight and suitable for downstream simulation tasks (such as physics simulation or driving simulation)?
2. How robust is the model to varying lighting conditions or the absence of panorama-derived illumination codes, and does it offer controllable rendering for different times of day?
3. Does the method generalize well to images outside the VIGOR dataset, including rural or non-urban areas?
4. It would be nice to see more analysis on the failure cases and the robustness of the method (e.g., non-planar surfaces, complex geometry, occlusion, etc.).

---

> ### Author Response · Authors · 2025-11-22
> **Responses to Reviewer Quvg (1/2)**
>
> ## W1: The paper lacks evaluation against metric 3D ground truth
>
> Following your advice, we have constructed a new benchmark with LiDAR-derived DSM data and have integrated quantitative metrics (MAE, RMSE) throughout our paper's experiments. This has strengthened our claims by providing direct, quantitative evidence.
>
> A detailed discussion of the new benchmark and a summary of the quantitative results can be found in the general response section at the beginning of our rebuttal, titled **"New Quantitative 3D Geometric Evaluation."**
>
> ---
>
> ## W2 & Q4: Failure mode and video metric
>
> **1. Discussion of Failure Modes:** You are correct that our method can face challenges with certain types of geometry. We have identified two primary failure modes:
>
> *   **Atypical Architectures:** Our generative model, by its nature, learns from the distribution of data it has seen. It struggles to reconstruct highly unusual or unique building shapes (e.g., "alien-like" architecture) because such examples are rare in the training data, and we lack explicit ground-truth shape information to enforce strong geometric constraints for them.
>
> *   **Non-Flat Terrain:** A assumption in our model is that the ground plane is locally flat. Learning terrain variations is extremely challenging given only sparse satellite and street-view imagery. Consequently, our model does not currently handle significant ground elevation changes like hills.
>
>
> We believe these limitations can be addressed in future work by incorporating multi-modal inputs. For instance, jointly learning from supplementary data like terrain maps or fine-grained building footprints could enable the generation of more realistic and diverse street-level 3D scenes.
>
> **2. Regarding Additional Metrics (Photometric Consistency & Temporal Flicker):** We appreciate the suggestion to use metrics like multi-view photometric consistency and temporal flicker. However, these metrics are not currently applicable to our evaluation setting due to the nature of the VIGOR dataset. VIGOR provides only a sparse collection of still panorama images for each scene, rather than dense, continuous video sequences. This data structure makes it impossible to measure temporal flicker. Similarly, the street-view images are too sparse to conduct a meaningful multi-view photometric consistency evaluation across a continuous camera path. Should denser video-based datasets for this task become available, adopting these metrics would be a high priority.
>
> We have strengthened our paper through the new Limitations section in Appendix Sec. G.
>
> ---
>
> ## W3 & Q3: Non-VIGOR test, handling of noisy or missing illumination inputs, methodological details and reproducibility.
>
> We address each of your points below.
>
> **1. Generalization to Non-VIGOR Data (e.g., rural areas):** This is an excellent point that we have also carefully considered. As detailed in our updated **Appendix Sec. F**, a direct evaluation on datasets like CVACT/CVUSA is unfortunately not feasible for two main technical reasons:
>
> *   **Technical Incompatibility:** Our 3D generation method requires multiview panoramic imagery per scene for geometric supervision. Datasets like CVACT and CVUSA provide only a single, centered ground-level image, which is insufficient for our training paradigm.
>
> *   **Unfair Domain Shift:** The satellite imagery in these datasets originates from different sources and has different visual characteristics (e.g., zoom levels) than VIGOR. Evaluating our VIGOR-trained model on them would unfairly conflate a test of geometric generalization with a test of resilience to a data-source domain shift.
>
>
> Instead, we test generalization using the **VIGOR-OOD split** (training on 3 cities, testing on 1 unseen city), which already introduces domain shifts in urban, suburban, and environmental styles. Our qualitative results (e.g., Figure 7) demonstrate that our method performs well in less-dense, suburban areas within VIGOR. We agree that scaling to more diverse geographies (e.g., rural, multi-continental) is a vital next step, which we highlight as a future direction.
>
> **2. Handling of Noisy or Missing Illumination Inputs:** For handling illumination, we follow the established practice of Sat2Density++. In cases where the sky is not visible in a panorama to extract illumination information, we use a fallback strategy and **set the illumination vector input to all zeros**. Our model demonstrates robustness in these scenarios. We acknowledge that we have not explicitly studied the effect of _noisy_ illumination inputs, which remains an interesting avenue for future investigation.
>
> **3. Reproducibility (Pretrained Models and Code):** **We will release our source code and pretrained model weights publicly upon the acceptance of the paper** to facilitate future research in this area. We have also updated our paper to include missing citations and improve diagram clarity as suggested.

---

> ### Author Response · Authors · 2025-11-22
> **Responses to Reviewer Quvg (2/2)**
>
> ## Q1: Are the generated meshes watertight and suitable for downstream simulation tasks (such as physics simulation or driving simulation)?
>
> Thank you for this important question. We investigated the watertightness of our generated meshes and found compelling results that also highlight the effectiveness of our proposed **Gravity-based Density Variation Loss (**$\mathcal{L}\_{\text{grav}}$**)**.
>
> In our analysis, we randomly sampled 100 generated meshes. We found that **approximately 95% of them are fully watertight**, as verified by the `is_watertight` check in the `trimesh` library. This high percentage demonstrates the general robustness of our method.
>
> Crucially, for comparison, we also analyzed meshes generated from a model trained **without** our Gravity-based Loss. In this ablation setting, the vast majority of meshes were **not watertight**, suffering from numerous holes, particularly in ground regions or areas with sparse satellite image coverage. This strongly suggests that our gravity loss plays a critical role in ensuring watertightness. It effectively regularizes the density field, encouraging it to form a solid, continuous ground plane and fill potential gaps, which is essential for producing a closed surface.
>
> We acknowledge that since our meshes are extracted from a learned NeRF representation via ray marching (more specifically, via Marching Cubes on the density grid), they are not natively modeled as manifold meshes from the start. Achieving 100% watertightness and optimizing them for specific downstream simulation tasks is a valuable direction. We plan to explore this further in future work, for instance, by investigating post-processing techniques or incorporating explicit manifold priors into our generation process.
>
> ---
>
> ## Q2:  How robust is the model to varying lighting conditions or the absence of panorama-derived illumination codes, and does it offer controllable rendering for different times of day?
>
> Thank you for this excellent question regarding lighting robustness and controllability. Our model's illumination module follows the established practice of Sat2Density++, and its robustness is confirmed during our evaluation where each satellite image is paired with a randomly sampled illumination code from the training set. This design allows for controllable rendering given an arbitrary illumination vector. However, direct control over the specific time of day (e.g., "render at 3 PM") is not currently possible. This is a limitation of the VIGOR dataset itself, which lacks the necessary time-of-day metadata for its panoramas, rather than a shortcoming of our model architecture. We consider enabling such time-specific control by exploring new datasets a promising direction for future research.

---

> ### Author Response · Authors · 2025-11-27
> **Welcome Further Feedbacks**
>
> Dear Reviewer Quvg,
>
> I hope you are well. With about five days left in the discussion period, we wanted to sincerely thank you for your practical and forward-thinking feedback.
>
> Your comments on evaluating with **metric 3D ground truth (DSM/LiDAR)** and your specific question about **watertight meshes** were especially insightful. Together, they pushed us to think critically about the bridge between our research and its real-world application. This led us to build a new geometry evaluation pipeline with real-world data, which not only allowed us to validate our method but also to provide a more concrete answer to your crucial question about suitability for downstream tasks.
>
> We hope our new results and detailed responses have fully addressed your concerns. If you have any further questions regarding this new evaluation or other points, we are more than ready to discuss.
>
> Since your feedback directly led to these significant improvements, we would be deeply grateful if this could be reflected in your final score. Thank you for your guidance in making our work more robust and application-oriented.

---

### Official Review · Reviewer_udS7 · 2025-11-01

**Soundness:** 2
**Presentation:** 3
**Contribution:** 2
**Rating:** 4
**Confidence:** 3

**Summary:**

This paper proposes a method for efficiently generating high-quality street-level 3D scenes from a single satellite image. Specifically, this method designs an end-to-end framework from satellite image to 3D. This framework first uses DINO v3 to encode the features of the input satellite image, and then decodes the encoded features into a Triplane feature field. Subsequently, volume rendering of this feature field is performed using an MLP to reconstruct the 3D scene. To effectively address potential issues such as edge artifacts, geometric distortions, and roof errors during scene reconstruction, this paper introduces various optimization strategies, including physical constraints and depth constraints. Experimental results show that, compared to existing methods such as Sat2Density++, Sat3DGen can generate street-level 3D scenes with more accurate geometric information and more detailed rendering results.

**Strengths:**

1. Well-written: The paper has a clear organizational structure, is well-written, and has a logical flow.
2. Targeted improvements to geometric stability: Gravity-based Density Variation Loss is proposed, which modulates the volume density along the direction of gravity, significantly alleviating the common "floating layers/holes" problem, making the reconstruction more coherent and more renderable.
3. Simultaneous improvement in rendering quality and cross-view consistency: Combining depth prior with multi-view supervision of panorama/perspective, covering a wider field of view and strengthening geometric constraints, resulting in more stable reconstruction and higher rendering fidelity, especially more reliable in details such as boundaries and roofs.

**Weaknesses:**

1. The contribution is unclear: Based on my understanding of this article, its basic framework is quite similar to Sat2Density++, with the core contribution being the introduction of depth conditions as training constraints. The authors need to better clarify the differences between this and the Sat2Density++ framework.

2. The experimental validation is insufficient. The authors only conducted experiments on VIGOR-OOD. To my knowledge, VIGOR-OOD is mainly designed for urban scene acquisition. For scenes that are more suburban or rural (e.g.,CVACT[1]), will the authors' method still have significant robustness?

3. Lack of evaluation baseline: As I understand it, based on the contribution of the paper, the focus is on optimizing the 3D geometry generated by the baseline method. Therefore, this paper should add a comparison with general Image-3D methods [2, 3].

[1] Liu, Liu, and Hongdong Li. "Lending orientation to neural networks for cross-view geo-localization." Proceedings of the IEEE/CVF conference on computer vision and pattern recognition. 2019.
[2] Xiang, Jianfeng, et al. "Structured 3d latents for scalable and versatile 3d generation." Proceedings of the Computer Vision and Pattern Recognition Conference. 2025.
[3] Hunyuan3D, Team, et al. "Hunyuan3D 2.1: From Images to High-Fidelity 3D Assets with Production-Ready PBR Material." arXiv preprint arXiv:2506.15442 (2025).

**Questions:**

1. Regarding the comparison of Figure 4(c), currently only Sat2Density is compared, excluding Sat2Density++. Given that Sat2Density++ clearly demonstrates a quality improvement over Sat2Density in its original paper, could you explain why this baseline was not included? Furthermore, since this paper achieves higher quantitative metrics, it is recommended to supplement the comparison with Sat2Density++ through parallel rendering, providing qualitative results under the same settings, to more comprehensively evaluate the differences between the two in terms of geometric consistency and texture fidelity.

---

> ### Author Response · Authors · 2025-11-22
> **Responses to Reviewer udS7 (1/2)**
>
> ## W1: The contribution is unclear. Based on my understanding of this article, its basic framework is quite similar to Sat2Density++, with the core contribution being the introduction of depth conditions as training constraints.  The authors need to better clarify the differences between this and the Sat2Density++ framework.
>
> While the depth prior is indeed a key component, our core contribution is a **holistic methodology** designed to fix the core geometric failures of the prior paradigm systematically. This methodology introduces a **synergistic suite of four distinct innovations**:
>
> 1.  The `Monocular Relative-Depth Prior`
>
> 2.  The `Gravity-based Density Variation Loss`
>
> 3.  The `Spatial Token` module
>
> 4.  The `Perspective View Training` strategy.
>
>
> Each of our components is designed to solve a specific geometric problem. The final leap in quality comes from all of them working together from different angles to improve the overall geometry. This comprehensive solution is the core contribution of our work.
>
> We realize our original manuscript did not highlight this holistic nature sufficiently. To address this thoroughly, we have prepared a detailed, centralized response at the beginning of our rebuttal, titled **"Clarification of Novelty and Detailed Comparison with Sat2Density++."** This section includes a point-by-point comparison table that we believe will fully clarify the fundamental differences and the significance of our work.
>
> ---
> ## W2: The experimental validation is insufficient, For scenes that are more suburban or rural (e.g.,CVACT\[1\]), will the authors' method still have significant robustness?
>
> Thank you for this insightful question regarding the scope of our experimental validation. We agree that evaluating the robustness of our method on diverse scenes is crucial. We address this concern by clarifying the scene types within VIGOR, explaining the technical unsuitability of CVACT/CVUSA for evaluation, and highlighting our existing generalization strategy. We have also updated Appendix F to reflect this detailed discussion.
>
> **1. Clarification on Scene Types (Urban vs. Suburban):** We would like to clarify the composition of these datasets. While datasets like VIGOR, CVACT, and CVUSA all consist of GPS-aligned satellite and ground-level image pairs, their ground-level data is sourced from Google Street View, which is captured by vehicles driving along road networks. Consequently, VIGOR itself contains a wide variety of scenes beyond dense urban cores. As shown in our qualitative results (e.g., Figure 7), our method is already evaluated on many less-dense, suburban-style areas with sparse buildings and more foliage. We argue that the primary challenge for generalization is not the urban/suburban distinction, but rather the significant domain shift in architectural styles and environments between different cities and continents.
>
> **2. Technical Unsuitability of CVACT/CVUSA for Evaluation:** You raised an excellent point about CVACT. As detailed in our updated Appendix Section F, these datasets are unsuitable for both training and, crucially for this discussion, fair OOD evaluation of our 3D generation method. The key reasons are:
>
> *   **Insufficient 3D Supervision for Training:** The single-panorama-centered-per-tile structure of CVACT/CVUSA provides extremely sparse supervision, which is incompatible with our method's requirement for multi-view ground-level imagery to learn reliable 3D geometry.
>
> *   **Unfair Domain Shift for Evaluation:** The satellite imagery in CVACT/CVUSA originates from different sources and has different zoom levels and visual appearances compared to VIGOR. Evaluating our VIGOR-trained model on this data would unfairly conflate a test of geometric robustness with a test of resilience to a data-source domain shift.
>
>
> **3. Our Existing Generalization Strategy (VIGOR-OOD):** We specifically designed our evaluation on the VIGOR-OOD test set to measure generalization. By training on three distinct US cities (Chicago, NYC, SF) and testing on an unseen city (Seattle), we are already evaluating our model's ability to handle significant domain gaps in city layouts and building styles. This setup provides a strong and relevant measure of generalization across different urban and suburban environments.
>
> **Future Direction:** We agree that scaling to more diverse geographies is a vital next step. We plan to address this in future work by curating a larger, multi-continental training dataset, which would enable a model to learn a truly global prior.
>
> In summary, while we did not use CVACT, this is due to its technical incompatibility for a fair evaluation of 3D geometry generation. Our use of the VIGOR-OOD split already provides a robust test of generalization across different city styles. We appreciate you raising this point, which has allowed us to clarify our methodology in the updated Appendix F.

---

> ### Author Response · Authors · 2025-11-22
> **Responses to Reviewer udS7 (2/2)**
>
> ## W3: Lack of geometry evaluation baseline:
>
> We thank the reviewer for pointing out these two important aspects of evaluation.
>
> 1.  **On Quantitative Geometry Evaluation (The Need for Metrics):** Regarding the general lack of quantitative evaluation, we fully agree this was a major limitation. To address this, a crucial point also raised by reviewers zCTK and Quvg, we have constructed a new benchmark by aligning high-resolution DSM ground truth with our test set. This allows us to provide quantitative validation (MAE, RMSE) in our main tables. We believe this new evaluation, detailed in the general response section **"New Quantitative 3D Geometric Evaluation,"** thoroughly addresses the concern about the lack of quantitative geometry tests.
>
> 2.  **On Comparing with general Image-to-3D methods:** We did not provide a direct, head-to-head comparison with methods like TRELLIS-S or Hunyuan-3D, because of a fundamental mismatch in both the task and the input data:
>
>     *   **Input Mismatch:** These methods are designed for object-centric images (e.g., a photo of a single car). A top-down, complex satellite image is an entirely different data modality for which these methods are not designed. A direct application would be ill-posed and likely fail to produce meaningful results.
>
>     *   **Task Mismatch:** They aim to generate a single, bounded object, whereas our goal is to generate a sprawling, holistic outdoor scene.
>
> ---
>
> ## Q1: Regarding the comparison in Figure 4(c)
>
> We sincerely thank the reviewer for their sharp observation and apologize for the confusing typo in the caption of Figure 4(c).
>
> The reviewer is absolutely right that the comparison should be against the stronger **Sat2Density++** baseline. This was indeed our intention and what we implemented, as stated in the "Qualitative Comparison" paragraph of our original submission: _"We compared the panorama video results generated by Sat2Density++ and our model."_
>
> Regrettably, the caption's typo contradicted our own text and caused this understandable confusion. We have now corrected the caption in the revised manuscript to accurately reflect the comparison against **Sat2Density++**.
>
> We appreciate the reviewer's careful reading, which has helped us resolve this inconsistency and improve the paper's clarity.

---

> ### Author Response · Authors · 2025-11-27
> **Welcome Further Feedbacks**
>
> Dear Reviewer udS7,
>
> I hope this message finds you well. As the discussion period progresses, we wanted to sincerely thank you for your critical and thought-provoking review.
>
> Your initial question regarding the "unclear contribution" was particularly impactful. It challenged us to better articulate the holistic nature of our methodology. This was the catalyst for us to synthesize our innovations into a cohesive framework, which we believe significantly strengthens the paper's core narrative.
>
> We hope this clearer presentation has fully addressed your primary concern. Should you have any further questions about our methodology or any other aspect of our work, we would be eager to provide more details.
>
> We would be very grateful if you might consider this significant improvement in your final assessment. Thank you again for pushing us to strengthen our core argument.

---

### Official Review · Reviewer_zCTK · 2025-11-02

**Soundness:** 3
**Presentation:** 3
**Contribution:** 3
**Rating:** 6
**Confidence:** 5

**Summary:**

Sat3DGen proposes a feed-forward framework to generate street-level 3D scenes from a single satellite image, addressing the trade-off between semantics and geometry in existing methods. Built on a tri-plane NeRF backbone, it introduces three key components: a gravity-based density variation loss to suppress floating artifacts and voids, spatial tokens to stabilize boundary geometry, and satellite-view depth regularization to resolve rooftop ambiguity. Additionally, it strengthens supervision by jointly training on panoramas and their projected perspective views. Experiments on VIGOR-OOD demonstrate superior performance and support downstream applications like DSM estimation and large-area mesh generation. The work’s core strength lies in targeted geometric optimizations, though it relies on combinations of existing techniques.

**Strengths:**

- **Clear framework design**: The pipeline (satellite encoding → tri-plane lifting → illumination-adaptive rendering) is logically coherent, with sufficient details for reproducibility.
- **effective geometric optimizations**: The gravity-based loss directly addresses volumetric field voids and floaters, which is a critical problem in scene-level NeRF-based generation, with clear qualitative and quantitative improvements.
- **Strong practical value**: Supports multiple downstream tasks (e.g. DSM estimation, multi-camera video generation, semantic-map-to-3D) without extra supervision, enhancing real-world applicability.
- **Comprehensive experimental validation**: Includes ablation studies for key components, cross-method comparisons, and qualitative/quantitative evaluations, ensuring result credibility.

**Weaknesses:**

- **relying on existing method combinations**: The core framework (tri-plane NeRF + 2D supervision) is borrowed from prior works (e.g., Sat2Density++). No breakthrough in methodology or framework design is presented.
- **Unclear motivation and lack of ablation for DINOv3 encoder**: The paper uses a frozen DINOv3 ViT encoder for satellite tokenization but provides no justification for choosing DINOv3 over other encoders. There is no ablation to verify whether DINOv3 contributes to performance gains, or if simpler encoders could achieve similar results, or if the model can generalize to out-of-distribution scenarios.
- **Lack of quantitative 3D geometric evaluation**: All geometric assessments are qualitative (mesh visualizations), with no quantitative metrics for 3D quality. This makes it hard to rigorously validate the claimed "superior geometric quality" compared to baselines like Sat2Density++.

**Questions:**

- What was the motivation for selecting DINOv3 as the satellite encoder? Have you conducted ablation studies comparing it with other encoders in terms of performance, computational cost, or token quality? If DINOv3 is replaced with a simpler encoder, how much performance degradation would occur?
- Could you supplement quantitative 3D metrics to objectively validate geometric improvements?

---

> ### Author Response · Authors · 2025-11-22
> **Responses to Reviewer zCTK**
>
> ### W1: relying on existing method combinations.
>
> Thank you for your comment. We agree that our model uses components from established works, and this was an intentional choice based on our research philosophy.
>
> We believe that building a successful deep learning model involves two **complementary and equally important parts**: a strong network architecture for its expressive power, and a set of well-designed training objectives to effectively guide the learning process. Our strategy was to build upon a solid foundation for the first part by using robust, general-purpose components (e.g., ViT and tri-plane). This allowed us to focus our novel contributions on the second part: designing a new suite of techniques specifically for learning better 3D geometry.
>
> Our key contributions，such as the **Gravity-based Density Variation Loss**, the **Monocular Relative-Depth Prior**, and our **Perspective View Training** strategy，are all designed to directly address fundamental 3D challenges like vertical plausibility, rooftop ambiguity, and sparse supervision.
>
> Our ablation studies confirm that these new geometry-focused techniques, working in concert with the solid backbone, are the primary driver of the performance gains. We have provided a more detailed summary of our novelty in the **"Clarification of Novelty..."** section of our general response.
>
> ---
> ### W2 & Q1: On the Choice of the DINOv3 Encoder.
>
> We thank the reviewer for this question. We have added a new ablation study to justify our choice.
>
> **1. Motivation: Following Best Practices for a Strong Encoder.**  our methodology **follows the successful practice of recent object-level image-to-3D generation methods**, which have demonstrated the immense value of powerful, pre-trained ViT backbones.
>
> We therefore adopt a frozen DINOv3 encoder. This allows us to leverage its state-of-the-art features at minimal training cost, ensuring our novel geometric contributions operate on a robust feature representation.
>
> **2. Ablation Study: Validating the Choice** To quantitatively validate this strategy, we conducted a new study comparing encoder choices under our "Full Model" setting:
>
> | Encoder | Trainable Params | Encoder Speed (FPS)↓ | $L _{dep}$ | $L _{grav}$ | Sp. Tok. | Per. Train | FID↓ |
> | --- | --- | --- | --- | --- | --- | --- | --- |
> | VAE-style (Trainable) | ~156M | 33 | ✓ | ✓ | ✓ | ✓ | 29.0 |
> | DINOv3-Base (Frozen) | ~76M | 30 | ✓ | ✓ | ✓ | ✓ | 23.0 |
> | **DINOv3-Large (Frozen)** | ~76M | 23 | ✓ | ✓ | ✓ | ✓ | **19.2** |
>
> **3. Conclusion** The results are clear:
>
> *   A simpler `VAE-style` encoder leads to a significant performance degradation (29.0 FID), validating that such encoders are a bottleneck for this task.
>
> *   Following the "best practice" of using a frozen `DINOv3` provides a massive quality boost (23.0 FID) while nearly halving the trainable parameters.
>
> *   Scaling to `DINOv3-Large` yields our best result (19.2 FID).
>
>
> This confirms that our strategic choice to adopt a strong, frozen backbone is a highly effective and efficient approach, which allows the true impact of our primary geometric contributions to be realized.
>
> Regarding **out-of-distribution (OOD) generalization**, our primary experimental setup on the VIGOR-OOD dataset is designed specifically for this purpose. We train our model on three distinct cities and evaluate it on a fourth, entirely held-out city. This cross-city evaluation serves as a challenging OOD scenario, as each city possesses unique architectural styles and layouts. Therefore, the strong performance on the unseen city directly demonstrates that our model learns a generalizable mapping from satellite to 3D structure, rather than simply overfitting to the patterns of the training data.
>
> ---
> ### W3 & Q2: Lack of quantitative 3D geometric evaluation:
>
> We thank the reviewer for this crucial suggestion. We fully agree on the need for quantitative 3D metrics. Following this advice, we undertook the significant effort of constructing a new benchmark by pairing the VIGOR-OOD test set with high-resolution DSM data.
>
> This has allowed us to add quantitative results to our paper, providing evidence for our method's geometric quality. For a detailed discussion of the challenges involved and a summary of the key findings, please see the general response section at the beginning of our rebuttal, titled "**New Quantitative 3D Geometric Evaluation.**"

---

> > ### Author Response · Authors · 2025-11-27
> > **Welcome Further Feedbacks**
> >
> > Dear Reviewer zCTK,
> >
> > I hope you are well. With about five days remaining in the discussion period, we wanted to follow up on your review.
> >
> > We especially want to thank you for pointing out the weakness regarding the "lack of quantitative 3D geometric evaluation." Your feedback was very helpful, as it pushed us to develop a geometry evaluation framework. This has significantly strengthened our paper by allowing us to scientifically support our claims about geometric quality.
> >
> > We hope our new results and detailed responses have successfully addressed this concern. If there is anything about these new evaluations, or any other point, that requires further clarification, please do not hesitate to let us know.
> >
> > We would be very grateful if you might consider these improvements in your final evaluation. Thank you again for your constructive review.

---

### Author Response · Authors · 2025-11-22

In addition to the clarifications above, we have responded to each reviewer individually. We are once again grateful for the time and effort they invested in helping us enhance this manuscript.

---

### Author Response · Authors · 2025-11-22
**Summary of Responses to Reviewers (4/4)**

## 2. Differentiating Sat3DGen from Sat2Density++

While we build upon the general feed-forward paradigm pioneered by works like Sat2Density++, our methodology leads to fundamental differences in design, training, and ultimately, capability.

| Feature Area | Sat2Density++ | **Sat3DGen (Ours)** |
| --- | --- | --- |
| **Design Philosophy** | Uses 3D as a proxy primarily for multi-view consistent video generation. | **Treats high-fidelity 3D geometry as the primary goal**, not just a proxy. |
| **Backbone** | VAE-style Encoder. | **Strong ViT backbone (DINOv3)**, a strategic choice informed by success in complex object-level generation. |
| **Geometric Strategy** | Minimal geometric constraints. | **Three geometry-focused strategy** (`Gravity`, `Depth`, `Spatial Token`) targeting specific 3D artifacts. |
| **Supervision Strategy** | Relies solely on sparse panoramic and satellite views. | **Introduces Perspective View Training**, a new training strategy to augment supervision fundamentally. |
| **Final Output Quality** | Produces coarse geometry (RMSE 6.76m) and poor photorealism (FID ~40). | **Achieves high-fidelity geometry (RMSE 5.20m) and photorealism (FID ~19)**, unlocking multiple downstream 3D applications. |

In summary, these differences illustrate our holistic, 'geometry-first' methodology. It is this systematic shift in design philosophy and strategy, not incremental tweaks, that directly leads to the state-of-the-art geometric and photorealistic results shown.

---

### Author Response · Authors · 2025-11-22
**Summary of Responses to Reviewers (3/4)**

# Clarification of Novelty and Detailed Comparison with Sat2Density++

## 1. Our Core Contribution: A Geometry-First Methodology

We sincerely thank Reviewers zCTK, udS7, and tZHZ for their critical feedback. Our initial manuscript presented our innovations as separate components, which understandably led to some concerns about novelty. We have revised the paper to better articulate our original design philosophy: a holistic, 'geometry-first' methodology, where each component is a targeted solution to a core geometric failure. This methodology is a direct and systematic answer to our crucial diagnosis: that prior methods fail due to the **extreme viewpoint gap** and **sparse, inconsistent supervision**.

Our primary innovation is an integrated suite of novel geometric constraints and training strategies, each targeting a specific failure mode identified in our analysis:

*   `Gravity-based Density Variation Loss` → Enforces plausible vertical structures.
*   `Spatial Token` → Regularizes peripheral layouts at boundary mismatches.
*   `Monocular Relative-Depth Prior` → Resolves rooftop ambiguity from a single overhead view.
*   `Perspective View Training` → Mitigates sparse supervision by increasing effective viewpoint coverage.

To address concerns (tZHZ) that our improvements might stem from the backbone rather than our methodology, we have **revised the presentation of our ablation study (Table 2)**. The new format now makes it explicit that the strong DINOv3 backbone is used consistently across **all** settings. This unequivocally demonstrates that the substantial performance leap (e.g., FID dropping from 35.6 to 19.2) is a direct result of our proposed synergistic components, not merely an artifact of a better backbone.

| Model | DINOv3 | $L _{dep}$ | $L _{grav}$ | Sp. Tok. | Per. Train | FID↓ | KID×100↓ | RMSE↓ |
| --- | --- | --- | --- | --- | --- | --- | --- | --- |
| Canonical Image-to-3D | ✓ |  |  |  |  | 35.6 | 30.1 | 6.21 |
| Base w/o $L _{dep}$| ✓ |  | ✓ | ✓ |  | 23.7 | 18.4 | 5.75 |
| Base w/o $L _{grav}$ | ✓ | ✓ |  | ✓ |  | 25.9 | 19.0 | 5.21 |
| Base w/o Spatial Tokens | ✓ | ✓ | ✓ |  |  | 24.8 | 18.1 | 5.64 |
| Base (Full w/o per. training) | ✓ | ✓ | ✓ | ✓ |  | 21.6 | 16.2 | 5.23 |
| Full model | ✓ | ✓ | ✓ | ✓ | ✓ | 19.2 | 13.6 | 5.20 |

---

### Author Response · Authors · 2025-11-22
**Summary of Responses to Reviewers (2/4)**

# New Quantitative 3D Geometric Evaluation (Response to zCTK, Quvg, and udS7)

We sincerely thank Reviewers **zCTK, Quvg, and udS7** for their feedback regarding the lack of quantitative geometric evaluation. We fully agree that this was a major limitation. Acting on this guidance, we have not only added the requested metrics but have also undertaken the significant effort to **construct a new benchmark** to make this evaluation possible.

**1. Contribution: Overcoming a Key Obstacle with a New Benchmark.** The primary obstacle to quantitative evaluation in this field is the abscent of public datasets that pair satellite images with high-resolution geometry. This is a far greater challenge than in the `object-level` domain. Overcoming this required a **non-trivial effort**:

*   **Locating Data:** We first had to identify and source suitable, high-resolution DSM data from specialized governmental archives that geographically overlapped with the satellite image collections.

*   **Enabling Alignment:** A precise alignment was only made possible because the VIGOR-OOD dataset, unlike many others, contains the **necessary georeferencing metadata**. The lack of such metadata often makes this type of quantitative evaluation infeasible for most datasets.


This process, detailed in the appendix **Section F**, represents a valuable contribution in itself, enabling geometric validation for future research.

**2. Impact: Definitive Quantitative Validation of Our Method.** This new benchmark allows us to present quantitative results. We provide a summary here to highlight the key findings. **The full, detailed versions of these tables can be found in the revised manuscript as Table 3 and Table 2, respectively.**

*   **Superiority over SOTA (Part of Table 3):** This summary compares our full model against the prior SOTA and a canonical baseline, demonstrating a dramatic improvement in geometric accuracy.


|  | MAE↓ | RMSE↓ | < 2.5 m ↑ | < 7.5 m ↑ |
| --- | --- | --- | --- | --- |
| Sat2Density++ | 4.72 | 6.76 | 49.69 | 83.65 |
| Canonical Image-to-3D | 4.23 | 6.21 | 52.73 | 84.54 |
| **Ours (Full model)** | **3.47** | **5.20** | **62.69** | **88.68** |

*   **Validation of Our Methodology (Part of Table 2):** To prove our proposed components are directly responsible for this gain, this summary of our ablation study shows the geometric impact (RMSE) of each component. It clearly demonstrates that our full model achieves the best performance.


|  | DINOv3 | $L _{dep}$ | $L _{grav}$ | Sp. Tok. | Per. Train | RMSE↓ |
| --- | --- | --- | --- | --- | --- | --- |
| Canonical Image-to-3D | ✓ |  |  |  |  | 6.21 |
| Base w/o $L _{dep}$ | ✓ |  | ✓ | ✓ |  | 5.75 |
| Base w/o $L _{grav}$ | ✓ | ✓ |  | ✓ |  | 5.21 |
| Base w/o Spatial Tokens | ✓ | ✓ | ✓ |  |  | 5.64 |
| Base (Full w/o per. training) | ✓ | ✓ | ✓ | ✓ |  | 5.23 |
| **Full model** | ✓ | ✓ | ✓ | ✓ | ✓ | **5.20** |

**3. Additional Materials & Final Word:** We have also included qualitative DSM visualizations in the appendix _Figure 9_. In summary, driven by the reviewers' feedback, we have **transformed a key weakness into a significant strength**. The new benchmark and the resulting quantitative analyses provide strong, unambiguous evidence for the geometric quality of our method. We are confident these additions have substantially improved the rigor and impact of our work.

---

### Author Response · Authors · 2025-11-22
**Summary of Responses to Reviewers (1/4)**

We thank all reviewers for their valuable feedback. We are grateful that they appreciate our clear framework (udS7, Quvg), effective geometric optimizations (zCTK, udS7, tZHZ), impressive performance (udS7, Quvg), and strong practical value (zCTK, Quvg).

Based on the reviewers' insightful suggestions, we have made substantial revisions to enhance the clarity, rigor, and scope of our work. All changes are marked in **blue** in the revised manuscript. The key updates are summarized below:

**Summary of Updates**

**1. Clarifying Our Core Contribution as a Holistic Methodology (zCTK, udS7, tZHZ):** Our initial manuscript presented our innovations as separate components, which understandably led to some concerns about novelty. We have revised the paper to better articulate our original design philosophy: a **holistic, 'geometry-first' methodology**, where each component is a targeted solution to a core geometric failure.

*   **Clarifying the Source of Improvement via Ablation Study:** To address concerns (tZHZ) that our improvements might stem from a stronger backbone rather than our methodology, we have revised the presentation of our ablation study (Table 2). The new format now explicitly indicates which components are active in each experiment, making it clear that the strong DINOv3 backbone is used consistently across **all** settings, including the weakest baseline. This unequivocally demonstrates that the substantial performance leap (e.g., FID dropping from 35.6 to 19.2) is a direct result of our proposed synergistic components, not merely an artifact of a better backbone.

**2. New Quantitative 3D Geometric Evaluation (zCTK, Quvg, udS7):** Spurred by the reviewers' crucial feedback, we now include a quantitative 3D evaluation. To achieve this, we undertook the non-trivial effort of **constructing a new benchmark** by sourcing and aligning high-resolution, LiDAR-derived DSM data with the VIGOR-OOD test set.

*   **Contribution:** We detail in the appendix **Section F** why this is a significant challenge (data scarcity, georeferencing requirements) that our work successfully overcomes, providing a valuable resource to the community.

*   **Impact:** This new benchmark allows us to report geometric results (such as MAE, RMSE) in the main paper, quantitatively validating our method's superiority.

**3. Enhanced Generalization and Robustness Discussion (udS7, Quvg):**
*   We have added a new discussion on the model's performance on non-urban scenes.
*   We have added a more detailed analysis of the model's failure modes and limitations.

**4. Correction of Figure Caption (udS7, tZHZ):**
*   We thank the reviewers for spotting this inconsistency. We have corrected the caption for Figure 4(c) to accurately state the baseline is Sat2Density++, matching the main text and eliminating any confusion.

---

### Author Response · Authors · 2025-11-28
**Subject: Seeking Feedback in Discussion Period**

Dear Reviewers and Area Chair,

As the Author-Reviewer-AC discussion period **nears its end in a few days**, we would like to gently inquire if there are **any concerns or unresolved questions** regarding our work.

The ICLR discussion process is designed to be an open, transparent, and interactive scientific dialogue. We feel this is **particularly crucial for a paper like ours, which has received a split decision (two positive and two negative reviews)**. A robust consensus can only be reached through the **deep engagement of every reviewer** in the discussion, leading to a fair and accurate decision.

We are deeply grateful for the reviewers' insightful feedback. Your comments on geometry quality (from Reviewers `zCTK`, `Quvg`, and `udS7`) were **instrumental, prompting us to introduce a new quantitative benchmark** for geometric evaluation along with corresponding experiments. We believe this is a **novel contribution that will benefit the community**. Similarly, the discussions on our contribution and model architecture (with Reviewers `zCTK`, `udS7`, and `tZHZ`) have enabled us to significantly refine and clarify our paper. We have treated all comments with the utmost respect and have incorporated corresponding revisions, which have **markedly improved the quality of our manuscript**.

We believe this dialogue-driven process is invaluable. We **kindly appeal to all reviewers to participate in the ongoing discussion**. We are more than willing to engage in a friendly and constructive discussion on any specific, objective points you may have and are very thankful for your selfless dedication of time and effort.

Finally, regarding the point raised by Reviewer `tZHZ` about the paper *"Controllable satellite-to-street-view synthesis with precise pose alignment and zero-shot environmental control. ICLR, 2025"* and its open-source status: We have reviewed the ICLR and arXiv PDFs of the paper, which contain no links to a code repository, and our subsequent searches using web engines did not yield an official implementation. We respect Reviewer `tZHZ`'s feedback and **commit to conducting a proper qualitative comparison should we be able to locate the official source code**.

Thank you once again for your invaluable time and effort.

---

### Meta-Review · Area_Chair_X1Mz · 2026-01-03

**Summary:**

I have carefully reviewed the comments from each reviewer and considered deeply about the split deicisons. Most reviewers appreciate the geometry-focused contributions and the effective experiments. Personally, I think this topic(sat2city) has great practical value and I look forward to wonderful work in this area.
The core concern remains that the distinction between this paper and Sat2Density++ has not been sufficiently clarified. While we acknowledge your proposed geometric improvement, the paper lacks adequate experiments and analysis to fully substantiate this differentiation. There should be more contents about it in the final version. While I believe this paper is worthy of acceptance, the aforementioned concerns should be addressed in the final version.

**Reviewer Concerns:**

Please see above.

**Reviewer Scores:**

The reviewer(tZHZ) confirmed that he is satisfied with the authors' responses to Q1, Q2, and Q3.", thus it is possible he raises the score to at least 4. Concerns raised by other reviewers are well addressed, and they are likely to maintain their positive scores. The most likely final score would be 6644. The core disagreement remains centered on the comparison with Sat2Density++. Although the discussion in the main text is severely insufficient, this experiment in Summary of Responses to Reviewers (2/4)) has demonstrated improvement. Therefore, I am inclined to adjust the overall score upward.

---

### Decision · Program_Chairs · 2026-01-26

Accept (Poster)